# Recent Progresses in Development of Biosensors for Thrombin Detection

**DOI:** 10.3390/bios12090767

**Published:** 2022-09-19

**Authors:** Reza Eivazzadeh-Keihan, Zahra Saadatidizaji, Ali Maleki, Miguel de la Guardia, Mohammad Mahdavi, Sajjad Barzegar, Samad Ahadian

**Affiliations:** 1Catalysts and Organic Synthesis Research Laboratory, Department of Chemistry, Iran University of Science and Technology, Tehran 16846-13114, Iran; 2Department of Analytical Chemistry, University of Valencia, Dr. Moliner 50, 46100 Valencia, Spain; 3Endocrinology and Metabolism Research Center, Endocrinology and Metabolism Clinical Sciences Institute, Tehran University of Medical Sciences, Tehran 14117-13139, Iran; 4Department of Chemical Engineering, School of Chemical and Petroleum Engineering, Shiraz University, Shiraz 71348-51154, Iran; 5NouBio, Los Angeles, CA 90025, USA

**Keywords:** thrombin, biosensor, aptamer, optical biosensor, electrochemical biosensor, detection

## Abstract

Thrombin is a serine protease with an essential role in homeostasis and blood coagulation. During vascular injuries, thrombin is generated from prothrombin, a plasma protein, to polymerize fibrinogen molecules into fibrin filaments. Moreover, thrombin is a potent stimulant for platelet activation, which causes blood clots to prevent bleeding. The rapid and sensitive detection of thrombin is important in biological analysis and clinical diagnosis. Hence, various biosensors for thrombin measurement have been developed. Biosensors are devices that produce a quantifiable signal from biological interactions in proportion to the concentration of a target analyte. An aptasensor is a biosensor in which a DNA or RNA aptamer has been used as a biological recognition element and can identify target molecules with a high degree of sensitivity and affinity. Designed biosensors could provide effective methods for the highly selective and specific detection of thrombin. This review has attempted to provide an update of the various biosensors proposed in the literature, which have been designed for thrombin detection. According to their various transducers, the constructions and compositions, the performance, benefits, and restrictions of each are summarized and compared.

## 1. Introduction

Rapid advancements in research have permitted the incorporation of biomaterials with adaptable sensors for the sustainable development of point-of-care systems to detect target compounds in different media [1]. Thus, a variety of sensors have been designed to identify various analytes (e.g., metal cations, abused drugs, neurotransmitters, circulating tumor cells [2], viruses [3], toxins [4], and cancer biomarkers [5]). A biosensor is a device that generally consists of three components, containing a biological element, a transducer, and a signal processor [6]. Biosensing technology has advanced in recent decades, with improvements in biorecognition, transducers, and signal processing. The biological element can be an enzyme, antibody, protein, whole cell, or aptamer. The transducer is the main component of the biosensors, and they can be classified into optical, electrochemical, piezoelectrical, calorimetric, or thermal categories, based on the physico-chemical characteristics of the transducer [7]. A biosensor can supply qualitative or quantitative information about biomolecular interactions by connecting a biological identification element to a transducer. Subsequently, the transducer converts the connection event into recognizable and quantifiable signals. An electrical instrument or an observer can be employed to monitor this signal [8]. When the biomolecular receptor consists of an aptamer, it is referred to as an aptasensor [9]. Indeed, the recognition element in aptasensors is a DNA or RNA aptamer [10]. Aptasensors are a kind of biosensor that can be used in a wide range of experimental conditions for diagnostics and drug delivery since they offer many advantages such as low-cost, quick response, easier modification, and easy merging within devices [11,12,13]. This can be achieved through the detection of small organic molecules and proteins [14]. Proteins are required by all of the body’s vital organs, and their prompt and accurate identification is essential in the diagnosis of various diseases [15]. The clinical methods for the detection of proteins are usually related to immunoassays, which contain some restrictions. Therefore, modern platforms for the detection of proteins have been developed. For this purpose, recently, aptamers and DNAzymes have been used for bioanalysis, nanotechnology, clinical diagnostics, and therapy [16]. Moreover, many 3D printing technologies have been recently utilized to construct biosensors or some parts of them due to the benefits of these approaches over conventional ones such as end-user customization, great functionality and adaptability, and quick prototyping. This technology has also quickly been developed in a wide range of biological and biomedical fields including organ on a-chip platforms, bioelectronics [17,18], microfluidic devices [19], and tissue-engineered implants [20].

Thrombin (TB) is a serine protease and has a key function in pathological and physiological coagulation, blood vessel hemostasis, and wound healing [21]. It is a special molecule that acts as both a procoagulant and an anticoagulant. TB controls its generation through activating coagulation factors V, VIII, and XI. When it comes to vascular injuries, it is generated from prothrombin in the wound area and induces the conversion of fibrinogen to fibrin. It also stimulates platelets, and through the activation of the factors XI and XIII, respectively, prevents fibrinolysis of the fibrin clots and affects the formation of a firm fibrin clot. This process, in turn, forms blood clots and, in such a way, prevents bleeding. The coagulation procedure begins when the concentration of TB in the blood reaches around 5–20 nM, and its concentration after the clot formation can be several hundred nM. The activity of TB as an anticoagulant is performed through binding to thrombomodulin (a membrane receptor protein), which begins a chain of actions that results in fibrinolysis [22,23]. The generation of TB is categorized into three sequential stages based on its final concentration and other relevant factors: (i) initiation (TB concentration is from 1 nM to 20–30 nM); (ii) propagation (from 20–30 nM up to more than 800 nM); and (iii) termination (when TB production stops and free thrombin is passive) [24]. Generally, the concentration of TB in the blood depends on the person’s physical condition. It may not be found in healthy people’s blood, while its level may vary from nM to µM during the coagulation cascade [25]. The imbalances in its concentration can lead to various diseases such as venous thromboembolism, nephrotic progress, central nervous system diseases, etc. In a specific case, pregnant women, who experience a physiological hypercoagulable state, are in peril of deep vein thrombosis and sets of momentous gestation complications such as preeclampsia, recurrent pregnancy loss, intrauterine growth restriction, or intrauterine fetal death [26]. Deep vein thrombosis is the generation of blood clots in the deep veins. Surgery, turbulence of blood flow, and endothelial injury are the factors that predispose to it, where surgery is the most common reason. It is a dangerous disorder that can cause morbidity and mortality [27,28]. Moreover, the alteration in the levels of TB is associated with leukemia, vascular wall inflammation, tumor growth, and metastasis [29]. As an illustration, it has a modulating effect on tumor cell proliferation by enhancing growth at a low concentration of cancer cells, preventing growth at a high concentration, and can also produce an apoptosis effect [30]. TB promotes tumor cell adherence to platelets, endothelial cells, and subendothelial matrix proteins. Platelet-mediated tumor cell sequestration shields tumor cells from immunologic host surveillance by preventing tumor cell eradication by natural killer cells, and so prolongs tumor survival in the circulation. Platelets also increase tumor cell growth and angiogenesis as well as metastasis [31].

In all of these cases, the rapid and accurate diagnosis of diseases can hinder their fatal consequences [32,33]. Hence, TB is employed as a main criterion for coagulation disorders, and accurate evaluation of its level assists in determining the course of diseases and developing treatment plans [34]. Consequently, providing highly sensitive and selective sensors for TB determination could be highly efficient in both research and clinical diagnosis [35]. It is worth noting that detecting TB in blood plasma is hugely challenging since its absorption on the sensor surface locally raises its concentration and can stimulate the coagulation cascade [22].

Thus far, several types of biosensors have been developed for detecting TB, namely enzyme-based, immunosensors, DNA-based biosensors, tissue-based, and thermal and piezoelectric biosensors. This comprehensive overview incorporates developments in biosensors for effective TB detection, which have been classified based on the transducer as optical, electrochemical, and other ones, and is the first systematic review that almost addresses all types of the recently proposed TB biosensors. The present literature update has considered the last five years of published literature and provides an outlook on the current technologies in TB biosensors, which will help acknowledge deficiencies in this field to be covered in future research.

## 2. Electrochemical Biosensors

Electrochemical biosensors have currently received a lot of consideration and are being used in a variety of fields due to their particular and excellent benefits in terms of affectability, easy production and use, cost, and miniaturization [36]. There have been significant advancements in clinical diagnosis, food quality regulation, and environmental monitoring [37]. Electrochemical techniques are favorable because of their great sensitivity, quick response, tiny sample consumption, and low cost [38]. Electrochemical biosensors are electrode-dependent sensors that measure the output of interactions of biomolecules with their targets on an electrode surface [39]. Typically, electrochemical sensing employs a transducer and three electrodes (counter electrode, working electrode, and reference electrode) [40]. Electrochemical transducers can measure the current, potential, conductance, or resistance and provide analyte concentration information in the form of electrical signals [41]. Various electrochemistry-driven biosensing techniques have recently been proposed for cost-effective and miniaturized analytical instruments based on amperometry, voltammetry, potentiometry, and electrochemical impedance spectroscopy (EIS). Although various electrochemical biosensors for TB measurement have been developed, there are still several constraints that must be resolved before their transfer into real-world applications. For instance, starting blood coagulation cascades by contact of the electrode as a foreign body with blood, which causes biofouling of the electrodes [42]. In addition, with regard to the presence of thrombin at nM quantities in the blood, there is a requirement for very sensitive biosensors for efficient detection. The utilization of nanoparticles and nanostructures can theoretically overcome these concerns and problems [38]. This review divides electrochemical TB biosensors into subsets of voltammetry, amperometry, and impedimetric ones.

### 2.1. Voltammetric Electrochemical Biosensors

Voltammetry is the measurement of the corresponding current of an electrochemical cell as a function of a time-reliant potential. Voltammetric techniques use a time-dependent changeable voltage with respect to the reference electrode to quantify the current response between the working electrode and the counter electrode in the region of the conducting of oxidation–reduction reactions. The three-electrode system is easily constructed on a single substrate and enables the measurement of a tiny amount of sample. Techniques such as square wave voltammetry (SWV), linear sweep voltammetry, stripping voltammetry, cyclic voltammetry, and differential pulse voltammetry (DPV) are used to develop electrochemical biosensors. They can analyze the magnitude of various molecules in environmental or biological samples [41,43,44]. One of the typical applications of voltammetric electrochemical biosensors in recent years has been the measurement of TB. Moreover, many signal amplification techniques have been commonly applied to these systems to promote their sensitivity [45]. The review of these efforts is presented in the following paragraphs.

A modified signal-on electrochemical aptasensor was developed based on the DNAzyme-driven DNA walker method for the rapid identification of TB. DNA walkers are nanoscale molecular systems composed of greatly programmable DNA self-assemble that can commutate chemical energy into mechanical, thus leading to signal amplification. In this aptasensor, the double operative hairpin DNA (HP) was designed with a substrate sequence of the Mg^2+^ dependent DNAzyme (to afford the DNA walkers’ driving force) and G-quadruplex forming fragment (to induce an impressively current response by binding hemin). The binding of TB with the aptamers created a sandwich construction when TB and DNA walker were added to the modified electrode. Then, the DNA walker could frequently bind and split HP with the aid of Mg^2+^ and unlock many active G-quadruplex sequences. Finally, hemin can connect to the G-quadruplex to construct G-quadruplex/hemin complexes, which leads to improved current efficiency measured through DPV [46]. The process of this DNA walker amplification strategy-based aptasensor is represented in Figure 1.

Another study fabricated an electrochemical aptasensor based on a tetrahedral DNA (T-DNA) probe to detect TB. The proposed structure helped control the density and orientation of the analyte; therefore, it could overcome the spatial effect. Moreover, this aptasensor used the hybridization chain reaction (HCR) to amplify the signal. The electrode substrate of the developed biosensor was reduced graphene oxide doped with sulfur and nitrogen. The co-doping process increased the electrode’s conductivity and particular surface area, which is desirable for expediting electron transfer and loading more DNA probes. Furthermore, gold nanoparticles (AuNPs) were electrode-posited; thus, the Au–S bonding modified the T-DNA probe on the electrode. The attachment of the aptamer to TB formed the aptamer target conjugates while the cDNA was preserved. Finally, the cDNAs activated HCR and led to high current response. In this process, hydroquinone was recycled with the assistance of H_2_O_2_ and caused an increase in the electrochemical reaction to avidin-HRP [15].

In 2020, Liang et al. established an efficient modulation of self-assembled DNA by enhanced proximity binding for the ultrasensitive electrochemical detection of TB. Three oligonucleotide sequences were used to self-assemble DNA (S1, S2, and Substrate). Poly adenine bound the substrate sequence to the gold particles on the electrode surface and regulated the distance between S1 and S2. One end of the probe sequences (S1, S2) was the TB aptamer (TBA), and the other end of S2 was loaded with methylene blue (MB) for signal indication. In the attendance of target molecules (herein TB), S1 and S2 formed G-quartets, which brought signal molecules closer to the sensor surface and increased electron transfer. Eventually, the redox current that represented TB concentration was measured by SWV. This biosensor could detect the TB concentration as low as 53.70 aM. Additionally, it provided a wide linear range from 1 fM to 100 nM [47].

Another technique that has been utilized to amplify TB detection was based on the host–guest chemistry between tetraferrocene (TTF) and ß-cyclodextrin (ß-CD). Ferrocene is a potential signal marker because of the excellent reversibility of the redox reaction, the production of a direct signal without the presence of other materials, simple process, and low cost. To benefit from this interaction, Huihui et al. synthetized TTF and deposited ß-CD on gold electrodes of an electrochemical aptasensor. In this system, TTF could not interact with the ß-CD when it was entrenched into the ends of TBA’s double stem-loop structure. However, this structure opened and changed to a special G-quarter form as the TB was presented to the medium. This process allowed the TTF molecules to enter the ß-CD cavity through host–guest recognition. Afterward, the TTF redox current increased eminently and could be observed by differential pulse voltammetry (DPV). This method provided the TB detection range of 4 pM–12.5 nM with a limit of 1.2 pM [12].

Metal-organic frameworks (MOFs) are novel porous materials of metal ions and organic ligands that are linked together by coordination bonds. Their advantageous characteristics such as their 3D infinite extension, clear pores, and exposed dynamic sites have promoted their application in biomimetic catalysis [48]. Especially in developing biosensors, Yu et al. developed the Ce (III, IV)-MOF electrocatalyst as a signal-amplifying label for an aptasensor. The MOF structure was conjugated with the proximity binding-induced DNA strand substitution and exonuclease III-assisted recycling process to amplify the detection of TB in complicated serum samples. In this system, the presence of TB allowed its selective connection to the proximity binding of two aptamers (S1, S2), which, in turn, caused the supplanting of S3. Then, the free S3 opened the capture probe 1 (CP1), resulting in the duplex DNAs. After that, exonuclease III took part in recycling S3; thus, the next loop started. This process generated many single-stranded receiver probes (CP1). Ultimately, the CP1 sequences bound to Au-Thi-Au@Ce (III, IV)-MOF labeled hairpin probe 1, causing a significant current (Figure 2) [49].

Another example of using MOFs in the construction of aptasensors was the application of the iron metal-organic frameworks (FeMOFs) as signal probes (SP) for TB detection. In this project, [Fe(CN)_6_]^3−/4−^ was used as an IR probe, and four DNA chains were annealed to fabricate DNA NTH with a high number of molecular modification localities, where the signal of Fe-MOFs can be observed in attendance with TB. The obtained aptasensor had easy performance and great sensitivity [50].

Another study employed Nickel-MOFs (Ni-MOFs) to develop an aptasensor for TB analysis. The Ni-MOFs were made up of the redox-active ligands 4,4′,4″-tricarboxytriphenylamine, the electroactive sources, and paramagnetic Ni^2+^ ions, nodes for electron transportation. The proposed method combined these electroactive materials with aptamers to detect TB. One aptamer (API) was adsorbed onto the electrode surface as a recipient aptamer, and the other one (APII) was tagged with a signal probe (APII/AuNPs/Ni-MOF) to provide a discrete detecting signal. The designed probe provided a sensitive electrochemical signal for TB due to the abundance of triphenylamine-based signal units ordered in a definite porous construction of the MOF [51].

In addition to MOFs, other metallic materials have been applied in biosensors to improve TB diagnosis. In 2020, an electrochemical aptasensor based on a dual signal amplification strategy of silver nanowires (AgNWs) and hollow Au-CeO_2_ was proposed to increase the efficiency of TB detection. AgNWs can be useful for binding biomolecules as well as in the improvement of electron transfer. Their combination with Au-CeO_2_ accelerates the reduction of H_2_O_2_ and improves the signal amplification of aptasensors. In this study, AgNWs were produced by using a polyol approach. Then, the manufactured AgNWs were deposed onto the indium tin oxide (ITO) electrode using a dipping process to firm amino functionalized TB capture AP1. Through the use of a sandwich-type construction, hollow Au-CeO_2_ was inserted on the electrode for settling sulfydryl functionalized TB reporter AP2 [52].

Researchers have also proposed other novel strategies to analyze TB in blood samples. For example, Zhao et al. designed an allosteric kissing complex-based electrochemical biosensor to detect TB. This method used the apical loop–loop or kissing interaction of the RNA–RNA base sequences to generate a kissing complex with two designed hairpins (immobilized HP1 and HP2 scaffold) on the electrode surface. After forming the kissing complex, an electrochemical response was produced by applying streptavidin-labeled alkaline phosphatase to the electrode. When TB, as the target protein, was bound to the recognition regions linked onto the HP2 scaffold, the HP2 stem unfolded due to the steric strain, which resulted in a reduction in the electrochemical signal relevant to protein quantification. In this way, the discrepancy between the electrochemical signal before and after TB binding could be used to efficiently evaluate its recognition [53].

An electrochemical aptasensor based on hyaluronic acid functionalized polydopamine (PDA) has been constructed for sensitive and low-fouling TB detection. In the construction of this aptasensor, 6-mercapto-1-hexanol (MCH), a low-cost and accessible small organic compound, was immobilized with PDA as a substrate. Hyaluronic acid with hydroxyl groups was grafted to the surface of the polydopamine modified electrode to improve the antifouling efficiency of the interface. Moreover, it was an excellent substrate for immobilizing TB-specific aptamers. Afterward, to analyze the analytical performance of the aptasensor, the DPV method was utilized to detect TB at various concentrations. The fabricated aptasensor demonstrated high sensitivity and selectivity with a detection limit of 0.03 pM and could detect thrombin in diluted human serum with significantly less non-specific adsorption [54].

Another creative strategy to develop an electrochemical biosensor was using efficient target-activated enzyme cascade electrocatalysis with a low background signal. When TB was presented to this biosensor, the horseradish peroxidase-mimicking DNAzyme (HRP mimicking DNAzyme) was produced as an electrochemical signal probe to initiate high-efficiency enzyme cascade electrocatalysis to reduce the background signal. Furthermore, for high enzyme cascade electrocatalytic efficiency, the detection sensitivity was increased further by regulating the side length of the rigid DNA tetrahedron scaffold anchored HRP-mimicking DNAzyme and GOx at adjacent vertices [55].

Carbon-based materials such as multi-walled carbon nanotubes (MWCNTs) and fullerene (C60) have been applied to construct biosensors. These materials render mechanical constancy, a great surface-to-volume ratio, low cost, electrocatalytic characteristics etc. [56,57,58,59,60]. Specifically, Jamei et al. fabricated an ultra-sensitive aptasensor based on polymer quantum dots and a C60/MWCNT polyethyleneimine (PEI) nanocomposite modified on the SPCE electrode surface to analyze TB (Figure 3). The presence of a high number of amine groups in the PEI composition assisted in the stabilization of biological molecules on the aptasensor. By connecting TB to the surface of the SPCE/C60/MWCNT PEI/PQdot/GLA/APT aptasensor, electron transfer was limited, and the analytical signal decreased in proportion to the TB concentration [59].

### 2.2. Impedimetric Electrochemical Biosensors

Electrochemical impedance spectroscopy (EIS) investigates the alternative current (AC) feedback with respect to frequency. In this technique, the electrochemical operation of the electrodes is ascertained by evaluating the EIS data based on alterations in current/potential at the specified frequencies, and as a reactive and inoffensive technology, have been broadly used in different fields [60]. One of these fields is the development of biosensors for detecting various analytes. This section presents the application of the EIS technique to develop TB detection probes.

Choosing an appropriate substrate material for electrodes used to develop an electrochemical aptasensor is critical. Self-polymerized polydopamine (PDA) film can act as a perfect electrode substrate material since it has a lot of hydrophilic hydroxyl groups (−OH) and can increase the material’s hydrophilicity. Furthermore, it can be used to immobilize biomolecules through the Michael addition reaction or Schiff base reactions and has less cytotoxicity than other systems. Xu et al. exploited this technique in their recent study. They proposed an effective electrochemical aptasensor for TB detection using the EIS method based on PDA film inserted with silver nanoparticles (AgNPs). PDA was self-polymerized onto the glassy carbon electrode (GCE), and the TB-aptamer was spliced onto the PDA-modified GCE via the Michael addition reaction. The addition of AgNPs enhances the film’s conductivity and improves the GCE’s surface area. MCH was employed to prevent non-target protein adsorption [61].

Another sensitive and selective aptasensor comprised of nanoporous anodized alumina arrays formed on a conductive ITO/glass electrode was designed to detect TB. Herein, the nanopores are consistently functionalized with a TBA and the EIS method was used to calculate the transient impedance of the alumina film-covered surface. The results showed that the sensing response is produced by modified ionic transport under the effect of steric hindrance and surface charge modification induced by ligand–receptor attachment between thrombin and the aptamer-covered alumina film. The constructed sensor detected TB with a detection limit of 10 pM and offered production simplicity, cost-effectiveness, and miniaturization [62].

Graphene oxide (GO), as a two-dimensional carbonaceous nanomaterial, has appropriate chemical and physical features such as a large surface area to volume ratio, high electrical conductivity, simple operation, ability to disperse evenly in pure water, etc. [63]. Therefore, an effective electrochemical impedance aptasensor, based on a AuNPs/dsDNA-GO nanocomposite, was designed for TB determination. In this system, as shown in Figure 4, GO was functionalized with dsDNA through an amidation reaction, and AuNPs were electrodeposited onto the dsDNA-GO to promote electrode efficiency. Ultimately, TBA captures TB molecules and causes an increase in the charge transfer resistance [64].

Silver nanoparticles (AgNPs) are frequently utilized in sensors and myriad technical applications. AgNPs can be employed in sensor technology due to electrical and thermal conductivity, biocompatibility, cost-effective, and antibacterial activity. As an example of their application in developing biosensors, a label-free electrochemical aptasensor based on AgNPs modified with graphite-like carbon nitride (g-C_3_N_4_) has been proposed for TB detection. The combination of g-C_3_N_4_ and AgNPs can be considered as a green photocatalyst. g-C_3_N_4_, a typical 2D π-coupled polymer semiconductor, provides a huge surface area, remarkable characteristics, and low cost. Thus, is beneficial in photocatalysts, degradation, and chemical sensors and can be modified with diverse metallic nanoparticles. In constructing an aptasensor based on these two technologies, Ag-g-C_3_N_4_ was prepared using NaBH_4_ as a reducing agent, and the g-C_3_N_4_ loaded with AgNPs was used to immobilize the aptamers. Then, NH_2_-terminated TB binding aptamers were fixed on the glass carbon electrode surface modified with Ag-g-C_3_N_4_. For electrochemical impedance signal enhancement in the determination of TB, this study used [Fe(CN)_6_]^3−/4−^ (as the electrochemical probe) [65].

HCR is another signal amplification strategy that has recently been considered in aptasensor signal enhancement due to its benefits such as simplicity, cost-effectiveness, enzyme-free, etc. This technique is used to design sensitive devices to diagnosis targets. In HCR, under optimal conditions, a hybridization occurrence between two stable DNA hairpins is induced by a DNA initiator strand to create a long nicked dsDNA formation. Accordingly, a label-free electrochemical aptasensor based on the intelligent combination of triple-helix DNA molecular switch (THMS) molecule and HCR signal amplification was proposed. It is worth noting that THMS-based sensing was obtained by altering the loop section sequence of the aptamer probe with consideration of Watson–Crick and Hoogsteen base pairings. The structure of the THMS system was comprised of the central aptamer, the aptamer sequence was flanked by two arm segments, and the triplex-forming oligonucleotide (TFO). In this strategy, molecular recognition occurs in a homogenous solution. The presence of the target caused the THMS to disassemble, which resulted in the release of TFO. Subsequently, the released TFO hybridized with the CP on the gold electrode and caused the production of dsDNA polymers via in situ HCR. Eventually, the signal amplification occurred with the electrostatic absorption of [Ru(NH_3_)_6_]^3+^ into the phosphate backbone of dsDNA polymers. This designed platform can be utilized to detect small molecules such ass TB [66].

An electrochemical aptasensor based on heparin-mimicking hyperbranched polyester nanoparticles with plentiful sulfonated acid molecules (HBPE-SO_3_ NPs) modified with GCE was built through the layer-by-layer self-assembly techniques for the identification of TB. The proposed aptasensor had a good binding tendency and selectivity for TB. The nano-sized influence and high density of ending units of hyperbranched polymers were significant in developing of this satisfactory aptasensor. In the presence of TB, the electron transfers at the electrode surface diminished due to the stereo-hindrance effect of the G-quadruplex–TB composites. This hindrance caused a noticeable reduction in response [67].

### 2.3. Amperometric Electrochemical Biosensors

Amperometric sensors measure the concentration of an analyte at a constant potential by measuring the current response. Herein, the amperometric method was employed in biosensors to measure the electrochemical characteristics of the prepared electrodes incubated with various TB concentrations. An electrochemical TB aptasensor was designed based on silver nanowire and particle (AgNWs and PCs) electrodes. Moreover, Pt-loaded hollow zinc ferrite nanospheres (Pt/ZnFe_2_O_4_) were used to amplify the signal. In this system, two aptamers and an ITO electrode (as the working electrode) were decorated with AgNWs and PCs to enhance the electrical efficiency. This electrode provided enough binding places for an aptamer (API). In addition, AgNWs and PCs had the catalytic activity to reduce H_2_O_2_, which encouraged signal amplification. Moreover, Pt/ZnFe_2_O_4_, due to high electrocatalytic activity for the reduction of H_2_O_2_, was used to stabilize the other aptamer (APII). The analytical efficiency of the proposed aptasensor was assessed by the aptasensor’s amperometric responses [68]. Another aptasensor similar to this strategy was built with Au nanoparticles ornamented graphene sheet (Au@GS) and CoPd dual nanoparticles (CoPd NPs). Au@GS was employed as a sensing platform for API immobility through the Au–S bond and CoPd NPs) as the labels of APII). This combination provided excellent catalytic properties for H_2_O_2_ reduction, which was helpful in increasing the aptasensor’s sensitivity. The electrochemical aptasensor was evaluated by measuring the current of electrocatalytic reduction toward H_2_O_2_. The proposed biosensor evidenced a linear range of 0.01–2.00 ng mL^−1^ and a low detection limit of 5 pg mL^−1^ with good selectivity, reproducibility, and stability [69].

A sensitive sandwich-type electrochemical aptasensor was built based on AuNP functionalized octahedral Cu_2_O nanocrystals. As illustrated in Figure 5, toluidine blue (Tb) and hemin/G-quadruplex tagged Cu_2_O-Au-BSA were used as signal labels. The electroactive Tb containing amino groups and amino-terminated TBA (as an electron transfer mediator) makes an Au–N bond with the Cu_2_O–Au nanocomposite interface. Subsequently, following the addition of H_2_O_2_ into the working buffer, the Cu_2_O, AuNPs, and hemin/G-quadruplex co-catalyzed the H_2_O_2_ reduction to encourage Tb electron transfer [70].

Table 1 summarizes the main characteristics and analytical features of electrochemical biosensors proposed in the literature for the determination of TB.

## 3. Optical Biosensors

Optical transformers reply to analytes following changes in optical characteristics such as absorption, reflectance, emission, etc. [96]. Indeed, the interaction of the radiation with a bioreceptor is used to accomplish optical detection in biosensors, and signal alterations are registered using a photodetector [97,98]. Various approaches have been presented to develop optical biosensors such as colorimetry, photoluminescence, chemiluminescence, photoelectrochemical, etc. 

### 3.1. Combined Colorimetric Optical Biosensors with Other Detectors

Among the different optical sensors, colorimetric assessment can give naked-eye output signal detection of the target molecules, which is beneficial for the fast recognition of real samples and involves advantages such as simplicity and easy operation. In this technique, the color of the solution is altered; therefore, it can be seen with the naked eye and through spectrophotometry. Nevertheless, optical biosensors restrict analytical operations because of their weak quantitative detection and sensitivity. However, integrating two detection techniques, colorimetry and electrochemistry, can compensate for these deficiencies. As a result, in a single target identification process, the addition of two types of reporter detectors and sensors can create two distinct types of signals and improve accuracy and applicability. Numerous signal probes are available for colorimetric and electrochemical sensors [99,100]. As a case study, a dual-channel approach for the determination of TB, based on a colorimetric aptasensor and a hydroxyapatite nanoparticle-based electrochemical sensor system as the reporter probe, has been proposed. When the TB aptamers were bound to the TB, the two reporter probes were split into two sections using magnetic separation. The electrochemical transformer section identified one component, and the colorimetric transformer section the other one. Moreover, to produce the electrochemical current, hydroxyapatite nanoparticles as a signal probe were mixed with magnetic nanoparticles. Meanwhile, in the attendance of TB, the oxidation of 3′3′5′5′-tetramethylbenzidine sulfate (TMB) was raised due to producing more signal probes and combining them with hemin to generate DNAzyme, which catalyzed the oxidation of TMB [99].

In another study, colorimetric and fluorometric techniques and AuNPs were used to produce dual signal readouts. In this project, three single-stranded DNA, AP29-I, AP29-II, and reporter probe (RP) were cleverly constructed. AP29-I and AP29-II are two split segments of the TB aptamer sequence that one long poly-adenine DNA sequence has attached to gold nanoparticles. Furthermore, the RP was entirely hybridized with a portion of AP29-II to produce duplex DNA, and the fluorescence of the fluorophore at one end was suppressed by the inner filter effect. When TB existed, the aptamer sequences interacted with it to form a secondary structure, which then caused AuNP aggregation events, and thereby, the solution’s color altered from wine red to purple. Concurrently, the report probes could be distinct from the AuNPs, and the fluorescence is recovered [100].

A label-free colorimetric aptasensor was constructed with the benefits of simplicity, efficiency, quick operation, and high sensitivity and selectivity, etc. In this protocol, the 29-bases long thrombin aptamer (TBA29) was employed as a TB ligand, and the 3,3′-diethylthiadicarbocyanine dye (DiSC_2_(5)) was used as a chromogenic substance. The DiSC_2_(5) can attach to the peptide nucleic acid (PNA)/DNA duplex through intercalation and causes a color alteration from blue to purple in the visible-light area without demanding any modification. In the presence of TB, TBA29 was linked to TB to create the G-quadruplex configuration, and the PNA was dissociated from the PNA/TBA29 duplex. Then, the solution changed from purple to blue. Eventually, the TB concentration could be measured using absorption spectroscopy or even the naked eye. Furthermore, this method can be utilized to detect other analytes by altering the sequences of the probe DNA for various targets [101].

### 3.2. Photoluminescence (PL) Optical Biosensors

PL is caused by an absorption/emission process between the fundamental and excited electronic energy levels in the molecular compounds and includes both the fluorescence and phosphorescence processes.

#### 3.2.1. Fluorescence Biosensors

Fluorescence-based transduction approaches have recently received significant attention in the construction of biosensors due to their high sensitivity, specificity, quick operation, suitability for distant analysis, reusability, etc. For the improvement in the performance of fluorescent biosensors, some factors such as analyte recognition units, signal transducers, and fluorescent labels should be thoroughly considered. A brighter fluorescent label and signal amplification techniques are typically used in the fabrication of biosensors to enhance the fluorescence signals [102].

Metal-enhanced fluorescence (MEF) is an event that happens while a fluorophore is placed at an optimal space (5–20 nm) from plasmonic nanostructures and causes an increase in quantum output and photo-stability and a reduction in fluorescence duration. Moreover, silver dendritic nanostructures (as MEF substrates), due to their effortless and economical procedure for synthesis, superstability, and large surface area, have been employed for sensing and recognition. A thrombin MEF-based aptasensor was assembled utilizing the thiolated 29-mer TBA as the receiving aptamer and the silver dendritic nanostructure as efficient MEF substrates, which considerably increased the fluorescence strength of the Cy5. The 15-mer thrombin-binding aptamer tagged with Cy5 was employed as a secondary aptamer. The designed aptasensor demonstrated great sensitivity and selectivity, with a detection limit of 32 pM. Moreover, due to its simplicity and low cost of construction, it is offered as a tool with significant potential for clinical disease diagnostics and for increasing the sensitivity of various technologies that use fluorescent dyes for analyte detection at trace amounts [103].

In another study, a non-label and enzyme-free fluorescence aptasensor was built using graphene oxide (GO) and a DNA strand with two segments (the first (S1) being a TB binding aptamer and the second (S2) being a G-quadruplex). In the presence of TB, a DNA strand was attached to the surface of the GO via π–π stacking. After adding N-methyl-mesoporphyrin IX (NMM), strong fluorescence was emitted due to the interaction between the G-quadruplex and NMM. The increment of fluorescence was linear in the TB concentration range of 0.37 M to 50 M. In the absence of TB, S remained fixed to the GO surface, and a low fluorescence was emitted. The fabricated aptasensor is very selective, inexpensive, uncomplicated, and presents potential uses in various disciplines [104].

Exo I, as a signal amplification component, and SYBR Green I (SGI) dye, a metrical cyanine dye, as a fluorescence signal enhancement device, were employed to build a label-free fluorescence aptasensor for the oversensitive determination of TB. In the presence of TB, the TB aptamer separates from its complementary DNA (cDNA), then forms an aptamer–thrombin complex with a G-quadruplex. In this way, an increase in the intensity of SGI fluorescence was observed. Moreover, by adding a small amount of Exo I, the G-quadruplex structure was broken, and the released TB bound the remaining dsDNA probe, which organized the reproduction cycle of this structure. As a result, the fluorescence signal of SGI and sensitivity of the detection strategy were enhanced. In contrast, in the absence of TB, the SGI was inserted into the dsDNA slots, as a result of which the SGI output provided a poor fluorescence signal [105]. Figure 6 depicts the overall performance of the mentioned method. 

Considering that the Fe_3_O_4_ magnetic nanoparticles have significant utilization in biosensing, a biosensing platform was designed for the detection of proteins with dye-labeled ssDNA and PEI-coated Fe_3_O_4_ nanoparticles. Polyethylenimine (PEI) is a cationic polymer that contains amines and has the ability to compress DNA. In the presence of Fe_3_O_4_ NPs, all of the dyes (HEX, FAM, TAMRA, etc.) tagged in DNA exhibited decreased fluorescence. By adding a target protein (here TB), binding between the dye-labeled DNA and the TB would modify the configuration of the dye-labeled DNA and disrupt the contacts between the dye-labeled ssDNA and Fe_3_O_4_ NPs. This process causes the release of dye-labeled DNA, so the fluorescence is almost recovered. Subsequently, the difference in fluorescence between the presence and absence of TB can be easily measured, which showed a linear association between the intensity of the FL and the added TB [106].

The majority of organic fluorescent dyes that are used as a signal transmission medium in optical aptasensors are toxic. As a consequence, DNA-stabilized metal nanoclusters (DNA-NCs) have recently been applied as a nascent substitute in fluorescent sensors for the identification of a variety of chemical and biological analytes. Furthermore, DNA-NCs have several benefits such as great stability and biocompatibility, less toxicity, very small size, high fluorescence emission, etc. Meanwhile, by varying the length and sequence of the DNA, the quantum yield and fluorescence emission of DNA-NCs can be changed. A simple aptasensor based on DNA-Cu/Ag NCs was developed for TB detection, wherein the DNA patterns were made up of a Cu/Ag NC-nucleation section at either end and a TBA section in the center. Due to the structural change of the aptamer in the attendance of TB, two DNA-Cu/Ag NCs approach close to each other, causing an increase in the fluorescence intensity of DNA-Cu/Ag NCs [107].

In a recent study, a new strategy was used to extend the duplex-specific nuclease (DSN) sensing utilization by using a TB RNA aptamer and a cRNA. In the presence of TB, the interaction of the RNA aptamer with TB resulted in the release of cRNA, which then hybridized with the fluorescent DNA probe. A fluorescent signal could be produced with the addition of DSN to this mixture. This approach showed a detection limit of 0.039 pM and great selectivity [108].

#### 3.2.2. Phosphorescence Biosensors

The room temperature phosphorescent detecting approach has recently been used in sensors. For instance, ratiometric phosphorescent probes [109], aggregation-induced phosphorescence enhancement [110], and the phosphorescent inner filter effect [111] have been explored. The transfer of energy for phosphorescence emission switches from the triplet state to the ground state, resulting in a prolonged lifespan and the prevention of autofluorescence and scattered light meddling from biological matrices. As a result, the signal-to-noise ratio of the phosphorescent biosensor is noticeably enhanced for measuring target analytes. As a case study, an “off–on” phosphorescent aptasensor was proposed to recognize TB based on phosphorescence resonance energy transfer. In this aptasensor, 3-mercaptopropionic acid capped Mn-doped ZnS quantum dots (MPA-Mn:ZnS QDs), as the energy grantors, and TBA labeled with BHQ2 dye (TBA-BHQ2), as the energy receivers, were used. In the presence of TB, a particular interaction between TBA-BHQ2 and TB occurs and leads to the establishment of a G-quadruplex structure, which causes a recuperation of phosphorescence. Eventually, the TB amount can be quantified by evaluating the restored phosphorescence intensity change value (ΔP). In general, the “off–on” phosphorescence aptasensor is utilized in studies to prevent mistaken positive signals induced by the nonspecific adsorption of fluorescent nanoprobes on the interface of non-target cells [112].

### 3.3. Chemiluminescence Optical Biosensors

Chemiluminescence (CL) is a light emission produced during chemical processes without an outer light origin or another energy source. In this process, the energy consumption through a chemical process brings about an unsteady, excited state. The emission of light occurs due to the subsequent transition of molecules to the ground state. The application of this method in the development of sensors provides, in general, a low limit of detection, a broad linear range, simplicity, effortless performance, and absence of background interference [113]. It has been applied in various fields including biochemistry [114], environmental evaluation [115], food safety [116], disease diagnosis [117], etc. 

A sensitive and selective chemiluminescence aptasensor for TB recognition was prepared using SiO_2_@GO@CF, a TBA as a recognition element, and a hemin/G-quadruplex DNAzyme (HG-DNAzyme) as a signal amplifier. In the structure of this aptasensor, the synthesized silica (SiO_2_) incorporates unique properties such as a vast and robust absorption surface area, high chemical purity, and excellent dispersion. GO has great chemical stability, minimal toxicity, biocompatibility, increased hydrophilicity. Associated carbon fiber (CF) also has high performance. TBA and HG-DNAzyme were sequentially modified at the interface of SiO_2_@GO@CF to produce the final sandwich rod-like construction composite (HG-DNAzyme/TBA/SiO_2_@GO@CF). In the presence of TB, HG-DNAzyme was released from the surface of TBA/SiO_2_@GO@CF and inserted into the sensing system, which catalyzed the luminol-H_2_O_2_ CL reaction [118].

Another chemiluminescence-based biosensor was constructed based on an ingenious integration of target-catalyzed hairpin assembly and Exo III-assisted signal amplification. In this study, a catalytic G-quadruplex-hemin DNAzyme was also used to enhance the formation of chemiluminescence in the presence of luminol and H_2_O_2_. The hairpin probes utilized in this work did not require any chemical modification, and the entire reaction could be performed in an isothermal aqueous solution. This approach resulted in high sensitivity for the determination of human TB with a detection limit of 0.92 pM. It could also detect various DNA-binding proteins by conjugating with different recognition units [119].

### 3.4. Electrochemiluminescence (ECL) Optical Biosensors

ECL is a light-emitting process induced by an intense electron-transfer interaction at the electrode surface between electrochemically produced molecules [120]. Because of its high controllability, minimal background signal, and simple optical setup, it provides a valuable analytic tool. By applying this technique, an aptasensor for TB determination with a detection limit as low as fM level was designed. As indicated in Figure 7, the tri-layer self-enhanced nanoparticles, as signal-amplification labels, and Au nanoparticles coated with ZnO nanorods (Au@ZnO NRs), were employed to modify the working electrode. The tri-layer self-enhanced nanoparticle (Au-PAMAM-Ru (II)-SnS_2_ NPs) consisted Ru (II)-doped SnS_2_ nanoparticles (Ru (II)-SnS_2_ NPs) as an inner-layer and polyamidoamine (PAMAM) dendrimers as a middle-layer and AuNPs as an external-layer. The AuNPs with strong electrocatalytic activity served as nanochannels for electron and energy transmission to increase the ECL signal. Meanwhile, the TBA1 was placed on the interface of Au@ZnO NRs on the working electrode, and TBA2 was labeled on the surface of the Au-PAMAM-Ru (II)-SnS_2_ NPs. The constructed aptsensing platform was combined with TB and Au-PAMAM-Ru(II)-SnS_2_ NPs labeled TBA 2 based on the sandwich-type protein-aptamer interactions. Finally, the ECL intensity was measured [121].

In another study, a label-free aptasensor for the recognition of TB was developed. In the construction of this aptasensor, palladium nanocones (Pd NCs), as an ECL emitter (fabricated by mercaptoethanol, amine-terminated polyamidoamine (PAMAM), Poly-L-lysine (PLL) as green protecting ligand), and a GCE were used. PAMAM was effectively fixed onto the electrode (GCE) through electro-grafting, and the Pd NCs were fabricated with mercaptoethanol by substituting PLL simultaneously and afterward fixed to the PAMAM/GCE. Eventually, with the addition of a specific TBA including the thiol group, and following the removal of nonspecific regions by bovine serum albumin, the ECL aptasensor was built. This highly sensitive and selective platform could measure TB with a lower detection limit down to 6.76 fM. [122].

Carbon dots (CDs) are luminous nanomaterials with unique characteristics such as low cytotoxicity, high biocompatibility, and durable chemical stability. CDs have been employed in a variety of fields. In most cases, CDs should be hybridized with other nanomaterials before being used in biosensing systems. The combination of CDs with nanosized gold materials, which have a high biocompatibility and simplicity of functionalization, render an appropriate strategy for developing ECL luminophores. You et al. benefited from this system in developing a sensitive ECL aptasensor centered on CD/AuNF nanohybrids to detect TB. Herein, the AuNPs in the center and CDs on the outside formed the CD/AuNF nanohybrids adjusted on a GCE. The CDs produced a strong ECL signal in the presence of H_2_O_2,_ and the aptasensor exhibited proper selectivity toward TB [123].

Another aptasensor was built based on the connection of zinc proto-porphyrin IX (ZnP) with aminated zeolitic imidazole framework-8 (ZIF-8) (represented as ZnP-NH-ZIF-8) to detect TB. The ZIF-8 is a common type of MOF that can be produced by coordinating the Zn ion with 2-methylimidazole ligands, resulting in a significant enhancement in catalytic activity due to the expanded surface area, numerous interior channels, configurable porosity, and tailorable activity. Subsequently, ZIF-8 was aminated via APTES thermal polymerization, followed by connecting with ZnP to produce a brighter ECL emitter. The ECL signals showed a 153-fold amplification for ZnP-NH-ZIF-8 due to the high catalytic kinetics for the oxygen reduction reaction. The developed aptasensor was examined for the detection of TB in dichloromethane with tetra-n butylammonium perchlorate as an electrolyte [124].

Another electrochemiluminescent aptasensor was designed for the detection of TB in human plasma specimens by stabilizing the three-dimensional nitrogen-doped graphene oxide (3D-NGO) on a GCE as a significant EC emitter. The disposed 3D-NGO exhibited advantageous characteristics including a wide surface area, high density and conductivity, less toxic character, good biocompatibility, and fast charge transition. The modified electrode was covered by graphene quantum dots (GQDs) and chitosan. Then, the NH_2_-aptamer was attached to the 3D-NGO/NGQD-chitosan through glutaric dialdehyde. The interaction of TB with the aptamer caused a reduction in ECL [125].

### 3.5. Photoelectrochemical (PEC) Optical Biosensors

PEC is a growing analytical technique for biological sensing focused on the correlation between variations in photovoltage/photocurrent and the concentration of the target analytes. PEC sensors have received a lot of attention regarding their high sensitivity, quick measuring, and low cost. The PEC technique has a smaller background signal than other traditional methods because the excitation signal (light) is entirely separated from the detection signal (current). Up to now, PEC biosensors have been employed for the determination of metal ions, cells, microRNA, and DNA. It is necessary to note that excellent photoactive materials with high photoelectric conversion efficiency have a significant impression on the analytical operation of these sensors [126,127]. 

Perylenetetracarboxylic acid (PTCA) was utilized in the construction of another photoelectrochemical aptasensor, in which the addition of C_60_@C_3_N_4_ nanocomposites, as a quencher on an electrode using a standard sandwich process, resulted in a considerably reduced photocurrent. Herein, C_3_N_4_ was used as a carrier for inactive fullerene (C_60_) loading to create C_60_@C_3_N_4_ nanocomposites, and AuNP coated PTCA were used as a sensing platform. The C_3_N_4_, with a large surface area, supplied huge binding sites for C_60_ stabilization. It also challenged PTCA in light absorption to produce a considerably low photocurrent in the attendance of the electron donor ascorbic acid. The fabricated aptasensor had a high sensitivity and performance and a low price. The steps of the mentioned process are represented in Figure 8 [128].

A Ag-TiO_2_ 3D nitrogen-doped graphene hydrogel (3DNGH) was synthesized through a simple hydrothermal process to measure TB. The photocurrent strength was dramatically improved in the developed label-free PEC aptasensor. The porous nature of 3DNGH provided a high available surface area, which assisted in the attachment of Ag and TiO_2_ nanoparticles. The TiO_2_ nanoparticle is a semiconductor material with long-run stability, no toxicity, high oxidizing activity, and excellent biocompatibility. Because of the poor absorption capability of TiO_2_ in visible light, graphene was combined with it to enhance the photoactivity performance. Moreover, the photoelectrochemical efficiency was further enhanced through the localized surface plasmon resonance (LSPR) of the AgNPs. Based on this nanocomposite, with increasing TB concentrations, the photocurrent was progressively reduced [126]. In another project, TiO_2_ was used to build another aptasensor, paired with CdS:Mn QDs, to increase the absorption spectrum and efficiency of the photoelectric conversion. The middle area of the QDs generated an electronic state that promoted electron transport by doping Mn^2+^. In this sandwich-type photoelectrochemical aptasensor, the precise identification of TB by the aptamer allowed for the gradual anchoring of the protein and signal aptamer on the electrode, bringing AuNPs away from the CdS:Mn/TiO_2_/ITO electrode and generating a high quenched photocurrent via energy transfer between the AuNPs and CdS:Mn QDs. The photocurrent reduced as the concentration of the target protein increased [129].

A photoelectrochemical aptasensor was designed using poly(5-formylindole)/Au (P5FIn/Au) nanocomposites for the detection of TB. Poly(5-formylindole) (P5FIn) provided high optical activity due to its low band gap, which makes it suitable for excitation by visible light. Meanwhile, the aldehyde groups of P5FIn promote the steady and effective stabilization of aptamers, therefore increasing their biological activity. Au enhances visible light absorption, and can also improve the photocurrent reaction, considering that it prevents the merging of P5FIn photo-generated electron-hole pairs [130]. 

### 3.6. Surface Plasmon Resonance (SPR) Optical Biosensors

SPR is another optical measuring method that does not demand the use of tagged proteins or fluorescent labels [131]. This technique was used in developing an antifouling polymer brush design and TBA bioreceptors to analyze TB. In this process, as observed in Figure 9, the TBA bioreceptors were effectively anchored on poly[(N-(2-hydroxypropyl)-methacrylamide)-co-(carboxybetaine methacrylamide)] brushes (HPMA-co-CBMAA), which was established on a gold sensor chip through photoinduced single-electron transfer living radical polymerization. Furthermore, to reduce sensor surface blockage by components of the blood samples, antifouling coverings such as self-assembled monolayers (SAMs) containing oligo ethylene glycol chains (OEG) were applied. OEG-SAMs are made up of closely packed requested molecules with short OEG headgroups toward the sample. Subsequently, the specific interaction of the aptamer bioreceptors with TB (and other compounds present in the blood such as HSA, IgG, and prothrombin) was analyzed, and variations in their functionality were reported when integrated into polymer brushes [132].

In another study, a biosensor based on SPR was developed for the detection of TB directly in human blood plasma. Indeed, the wavelength spectroscopy of diffraction-coupled surface plasmons on a chip with a periodically corrugated gold film coated with an antifouling thin polymer layer composed of poly(HPMA-co-CBMAA) brushes was the foundation of this biosensor platform. Aptamer ligands were covalently immobilized using the carboxylate groups combined with the polymer brushes. These ligands were chosen to selectively collect the TB analyte from the tested blood plasma sample while not activating the coagulatory process at the biosensor surface with poly(HPMA-co-CBMAA) brushes [22].

### 3.7. Waveguide Optical Biosensors

The sensing plate utilized in the waveguide-mode sensor contains a reflecting coating and a lucid dielectric waveguide on the substrate glass. When the light gets in through the prism, it is associated with a waveguide, spreading the incident light at a specific angle. The strength of the reflected light falls substantially if light arrives at an angle similar to this particular angle. A waveguide-mode sensor detects molecule absorption by measuring variations in the amount of reflected light. Waveguide-mode sensors can be used to analyze various biomolecular interactions, providing physical and chemical stability and high susceptibility [133].

Light is restricted in a leaky waveguide (LW) by processes other than total internal reflection (TIR) at one or both waveguide interfaces. The resonance angle of LWs can be seen by placing a metal layer, immobilizing an appropriate dye in the waveguide, or constructing strips of waveguides. The resonance angle is ascertained by the waveguide’s effective refractive index, which can change due to the analyte adhering to bioreceptors fixed in the waveguide, resulting in the biosensing process. A leaky waveguide aptasensor was designed for measuring TB, in which aptamers were anchored in a waveguiding film made of mesoporous chitosan. Aptamers attach to their targets specifically via hydrogen bonding, electrostatic, and hydrophobic interactions. The presented aptasensor had high durability, low cost, and a simple manufacturing procedure [134].

An optical aptasensor based on photonic crystal (PC) sensing devices was created to measure rising TB concentrations, with a limit of detection of 33.5 pmol/L, a linear range at nmol/L level, and a response time of about 2 min. The chosen PC-sensing device was made up of a silicon 1D periodic corrugated waveguide. In its construction, the photonic band gap in the transmission spectrum was altered whenever the target TB molecules were fixed on its surface. [24].

Optical fibers operate as light transducers and provide a great substrate for attaching receivers and accomplishing fast and label-free biomolecule identification. They contain several satisfactory properties such as low cost, insignificant size, low weight, magnetic resonance compatibility, and remote and multiplexed sensing abilities. Light is restricted to the center of the optical fiber and is transferred with minimal propagation loss via complete internal reflection. Recently, optical fiber sensors have been regarded as a high-performance tool to identify various analytes. The fiber construction was altered for sensing purposes to enable light interaction with the adjacent areas at the sensing sites. Chemical etching, tapering, polishing, bending, and photo-inscription are examples of such alterations. Between modified fibers, grating-based optical fiber sensors such as tilted fiber Bragg gratings (TFBGs) have been widely used for biomolecule recognition. TFBG sensors offer easy and efficient platforms for bioreceptor insertion and measurement. A sensitive fiber aptasensor, based on etched tilted fiber Bragg grating, was built by combining two fiber alterations (namely fiber grating photo-inscription and chemical etching) for selective TB measurement. Simple surface chemistry centered on silane-coupling agents was employed directly on the fiber coating for the stabilization of bioreceptors. Due to the greatly enhanced etching procedure, the overall fiber diameter was reduced from 125 μm to 13 μm, and the sensitivity of the TFBG fiber enhanced dramatically, reaching 23.38 nm/RIU [135].

In another study, a label-free TB-sensing aptasensor was proposed based on a tapered fiber-optic interferometer. This biosensor was simply manufactured by tapering a commercial double-coating fiber down to 5 µm in diameter. The detection apparatus was dependent on RI measurement. The sensitivity was calculated by tracking the wavelength shift of a chosen resonance in the emission spectrum against the variation in RI. For the efficient targeted detection of TB, the interface of the tapered fiber was conjugated with TBA [136].

### 3.8. Other Optical Biosensors

Resonance light scattering (RLS) is frequently utilized in various investigations due to its great sensitivity, simplicity, and quickness. In this technique, the incident light energy is within or near the system’s absorption band. Light scattering signals are easily obtained by simultaneously monitoring both monochromators, the excitation and emission, on a common spectrofluorometer [137,138]. An aptasensor was proposed for measuring TB using the RLS method and magnetic nanoparticles (MNPs) as the probe. Streptavidin-coated MNPs were attached to biotin-labeled TBA. In the presence of TB on the surface of MNPs, a sandwich structure developed and caused MNP accumulation. MNPs have good biocompatibility and capacity for protein loading, and their magnetism simplifies the sensor building procedures significantly. Herein, the fabricated aptasensor could be utilized frequently by heating the solution and thereby wrecking the sandwich construction of the TBA and then accumulating the MNPs by applying an external magnetic field. Furthermore, the overall process of analyte identification can be completed in a matter of a few minutes [139].

Resonance Rayleigh scattering (RRS) is an elastic light scattering created by placing the excitation wavelength close to the absorption spectrum of a scattering molecule. As a result, its strength is considerably greater than the usual light scattering strength. This method is used in analyses because of its speed, simplicity of equipment, high sensitivity, size, and charge dispersion. This technique has been used with various probes including AuNPs, dyes, and graphite oxide. For instance, an RRS aptasensor was proposed for the measurement of TB by incorporating in situ generated and inserted AuNPs into a polyvinyl alcohol–borax hydrogel (PBH) without using any reducing agent and then connecting a thiolated-TBA to the interface of AuNPs. The variation in RRS intensity in the presence and absence of TB was computed and chosen as the monitoring signal [140].

In another study, a label-free liquid crystal (LC) aptasensor was designed to determine TB with a low detection limit of 1 pg/mL (26.7 fM), and a linear range of 26.7 fM to 26.7 μM. LC-based approaches have potential benefits over traditional techniques since they do not require complex apparatus or labeling, for instance, enzymatic or fluorescent tagging. The particular binding of TB to the fixed TBA on the glass slide’s surface disturbs the surface topography. It causes the orientational change of LCs from regular to randomized, leading to optical modifications from dark to bright. Overall, this aptasensor provides a sensitive, selective, quick, and economical technique [35].

A lateral flow aptasensor (LFA) was developed to identify platelet-derived growth factor-BB (PDGF-BB) and TB at the same time in a complex biological matrix. The LFA technique has been used to identify a variety of targets because of its extremely economical and adaptable properties. Thiolated aptamers were attached to the surface of AuNP, and biotinylated aptamers were established on the test zones of an LFA nitrocellulose membrane. AuNPs were collected on the LFA’s test zones and given red bands to allow for visual identification of the intended proteins. Subsequently, the test band intensity levels were measured with a portable strip detector employing the ‘Gold-Bio strip reader’ software to obtain the quantitative data [141].

The integration of the enzyme-linked immunosorbent assay (ELISA) dual affinity reagent sandwich design with surface enhanced Raman spectroscopy (SERS) was reported for quantitative detection systems. The SERS-based method used a pair of affinity reagents, one connected to AuNPs and the other attached to a gold film, each tagged with a Raman sensor. Signals obtained by both affinity labels confirmed the presence of TB. However, signals produced by nonspecific binding lacked one or both tags and allowed us to differentiate genuine from false positives quickly. The Aunt–Au film sandwich generated in the attendance of the target molecule produced a SERS hot spot, resulting in a powerful and clear signal, indicating a genuine positive. Sites without any responses from every reporter were discarded as providing no signal, and sites with just a single reporter’s participation were dismissed as false positives. Moreover, to analyze the results and minimize false positives, data were examined through CLS analysis. Eventually, this SERS-linked sandwich assay exhibited high selectivity for the recognition of human α-thrombin, and can discriminate between genuine and misleading bindings [142]. Another aptasensor based on the SERS approach was designed for the recognition of TB using dimeric AuNPs and magnetic nanoparticles. Moreover, the reagent 1,2-bis (4-pyridyl) ethylene (BPE) on the AuNP dimers acted as a coupling agent and as a Raman reporter. In this procedure, to create a complete monitoring device, the synthesized AuNP-dimers were loaded with the complement aptamer and hybridized with the TBA-modified MNP. When TB was present, the double-strand established between the AuNP-dimers and MNP would be unraveled, and the Raman signals would be enhanced by the use of magnetic separation [143].

Table 2 summarizes the main aspects and analytical features of different optical biosensors proposed in the literature for the determination of TB.

## 4. Other Biosensors

Capacitive sensors provide various applications in a broad range of fields due to their advantages including reasonable prices, quick response, non-invasiveness, and electrode design flexibility. A capacitive biosensor can be built by establishing bio-recognition components and detecting changes in the dielectric characteristics when an analyte engages the bioreceptor. An alteration in the dielectric characteristics of the material between a pair of electrodes causes a change in capacitance, which is associated with the number of bound molecules and bio-recognition components at the electrode interface. The presence of an insulating layer on the electrodes is also another significant challenge in the construction of capacitive sensors. If the insulating layer is inadequately prepared, Faradic currents greatly reduce the capacitive signal. Furthermore, it is also absolutely essential to pay attention to the insulating layer’s material and thickness, which ought to be as thin as feasible and can be used as self-assembled monolayers (SAM), a single layer of molecules on a substrate that has a high level of orientation, molecular organization, and packing, for this purpose. Inspired by these principles, a label-free aptasensor was designed to measure TB. In this system, the electrode pair was made up of a gold one, produced from compact discs and an ITO film, as the opposite electrode. Aptamers (as bio-recognition factors) and 1-dodecanethiol (for preventing non-target protein binding) were utilized to create a self-assembled monolayer (SAM) on the gold electrode surfaces. The ultra-small capacitance changes caused by TB binding with the aptamers were detected using a home-made capacitance measurement circuit based on the switched capacitor, and capacitance detection were accomplished by directly assessing the storage charge at a constant voltage [173].

Quartz crystal microbalance (QCM) as a detection method can be used in the construction of aptasensors. The detecting concept of the QCM biosensor system is based on the Sauerbrey relation, which states that the decrease in frequency variation is linearly proportional to the added mass on the crystal. Accordingly, in a study, by incorporating magnetic aptamer-microbeads, an aptasensor was built for the rapid identification of TB. As can be seen in Figure 10, to fabricate this aptasensor, the TBA was fixed on magnetic poly(2-hydroxyethyl methacrylate-ethylene glycol dimethacrylate-vinylene carbonate), Mp (HEMA-EGDMA-VC), microbeads through a ring-opening reaction. The Mp (HEMA-EGDMA-VC)-TBA microbeads efficiently absorbed TB from serum in a short time, and the Mp (HEMA-EGDMA-VC)-TBA eluted protein was transported to the QCM aptasensor, which recognized TB from serum. Subsequently, the surface of gold-plated QCM crystals was covered with L-cysteine, and a TBA was fixed on the interface of the L-cysteine-coated QCM crystals through a glutaraldehyde connection [174].

One of the most basic equipment and facilities in a laboratory is the electronic balance, which is used to calculate the mass of the sample with great precision accurately. Based on this equipment, a simple and effective aptasensor was proposed, in which two aptamers were employed: one was fixed on the surface of magnetic microparticles, and the other was loaded with platinum nanoparticles (PtNPs). The captured PtNPs catalyzed the decomposition of H_2_O_2_. Thus, it generated a substantial amount of H_2_O and O_2_. This process transferred a particular volume of water in a drainage device because the pressure inside the vial was higher than the level outside the vial. Despite the huge increase in gas pressure of the closed system, the molecular identification could be converted into a pressure signal. As the TB concentration increased in the sample, the weight of water also rose [175].

In another study, a barometer-based biosensor was designed to detect carcinoembryonic antigen, ractopamine, TB in serum, and Hg^2+^ in river water. In constructing this biosensor, core/shell Au@PtNPs were used as probes, which exhibited high catalytic stability and activity in the decomposition of H_2_O_2_ to O_2_. The produced O_2_ was concentrated within the limited chamber and, as a result, led to a rise in pressure, which was measured using a barometer. Furthermore, to facilitate the interpretation of data, a smartphone app was designed that evaluates, saves, and transmits data wirelessly [176].

## 5. Future Trends and Conclusions

Here, we presented a review of biosensors for thrombin detection. Thrombin is a serine protease that is essential for hemostasis, and coagulation abnormalities can induce various pathologies. For instance, imbalance in its level in the blood causes a wide range of abnormalities including cancer, venous thromboembolic diseases, nephrotic progression, nervous system disorders, chronic inflammatory diseases, and other diseases. Therefore, there is a considerable demand for the quick and low-cost measurement of thrombin. Currently, biosensors are widely employed in biomedical diagnostics. The wide application of biosensors is due to their great potential for the specific, rapid, and sensitive detection of target analytes. The literature update evidences the tremendous development made in the search for fast, sensitive, and accurate thrombin determination by using biosensors. Although numerous biosensors have been introduced to date for the detection of thrombin, there are several challenges involved in their advancement. Some of these challenges include: (i) improving the transduction process; (ii) biofouling of the electrodes; (iii) biosensor miniaturization, in which miniaturized nano-aptasensors can be sophisticated sensing platforms due to the development of microfluidic and on-chip-based techniques; (iv) increasing the transducer performance by improving sensitivity, selectivity, reproducibility, shortening response time, etc.; and (v) the presence of thrombin at nM levels and the alteration of its content in the human body by physiological processes shows a demand for the development of highly sensitive biosensors to monitor thrombin content in vivo. With regard to experience, these challenges can be overcome by combining sensor technologies with nanomaterials, and it is noteworthy that optimal performance has been attained by the combination of multi-material components. Future studies need to concentrate on integrating functional nanomaterials to enhance the synergistic effects and therefore improve the sensor’s overall performance. Meanwhile, increasing the characteristics of the proposed biosensors, together with improving the biosensor’s surface with various detection elements, can be employed for the simultaneous determination of TB and other important proteins in physiological samples. Nowadays, there are many strands to selectively recognize thrombin and the different studies have provided alternatives for the use of different quantification procedures to avoid matrix interferences. However, additional efforts must be made to move from the bench to the real-world point-of-care use of thrombin biosensors in diagnostics.

## Figures and Tables

**Figure 1 biosensors-12-00767-f001:**
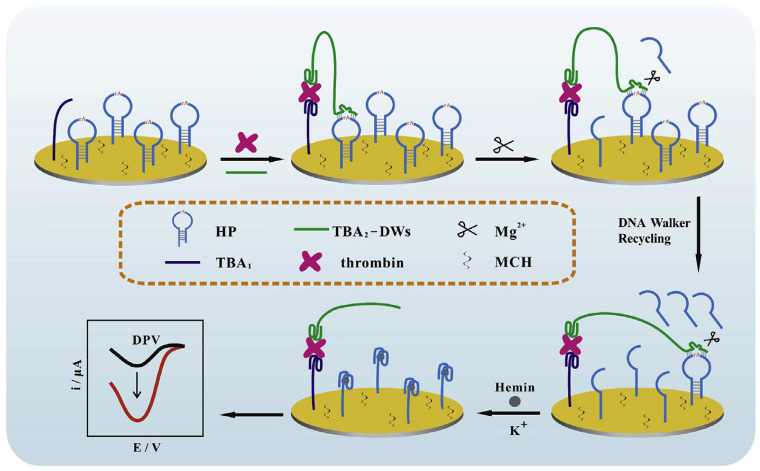
An electrochemical aptasensor based on the DNA walker amplification strategy for TB determination [46].

**Figure 2 biosensors-12-00767-f002:**
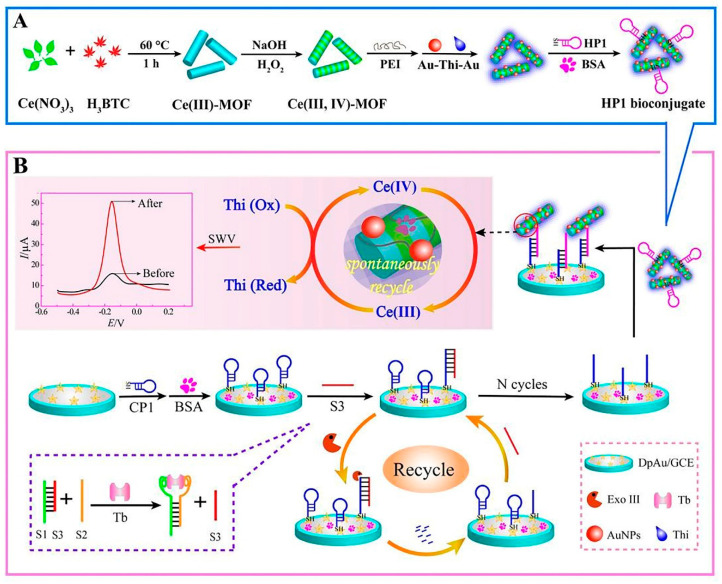
MOF-voltammetric aptasensor for TB determination: (**A**) Construction steps of HP1 bioconjugate (HP1/Au-Thi-Au@Ce (III, IV)-MOF), and (**B**) the manufacturing process of the Ce (III, IV)-MOF electrocatalyst with Exo III-assisted recycling amplification [49].

**Figure 3 biosensors-12-00767-f003:**
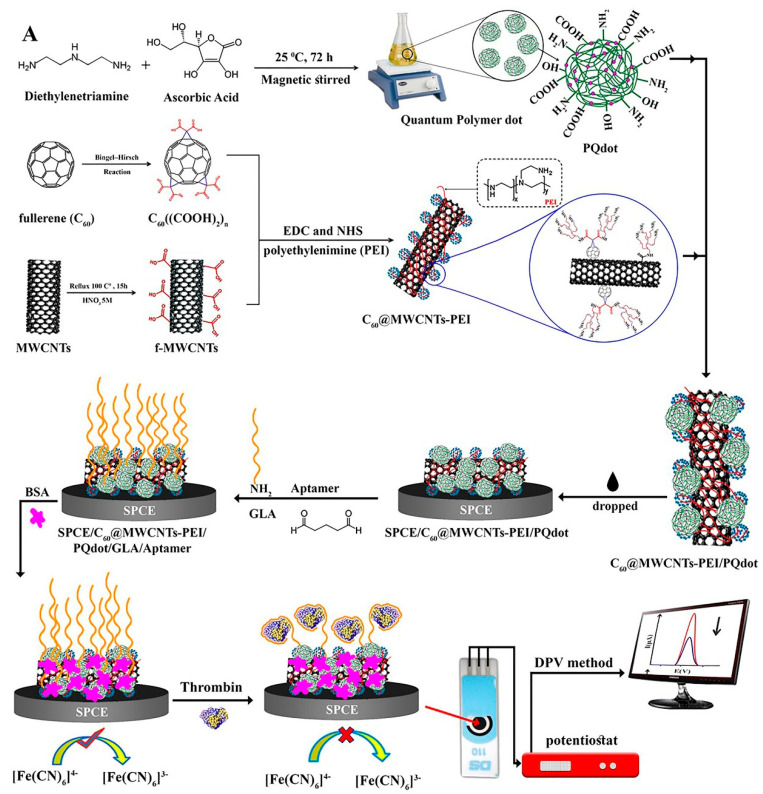
Multi-walled carbon nanotubes and fullerene-based electrochemical aptasensor fabrication for TB detection using the DPV method, and C60/MWCNTsPEI/PQdot [59].

**Figure 4 biosensors-12-00767-f004:**
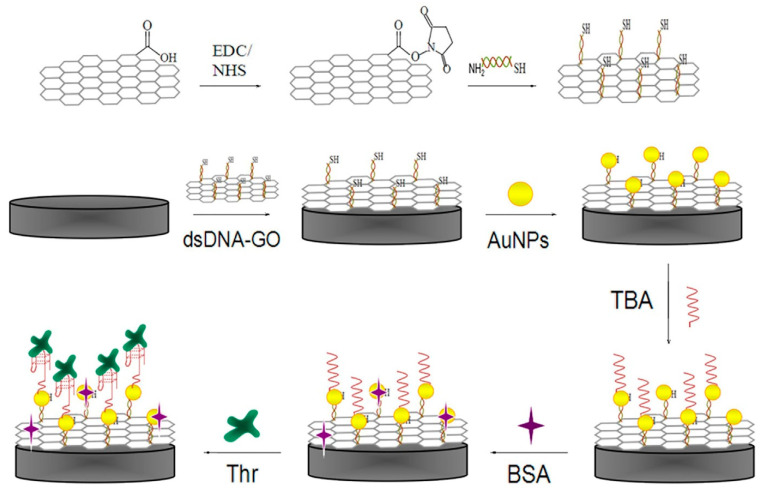
Step-wise formation of an impedance aptasensor for TB determination based on the AuNPs/dsDNA-GO nanocomposite [64].

**Figure 5 biosensors-12-00767-f005:**
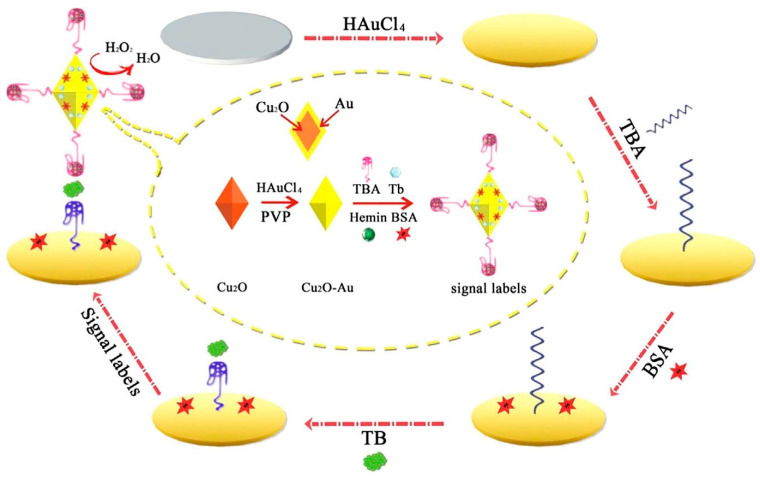
The signal amplification strategy for a chronoamperometry-electrochemical aptasensor for the determination of TB based on the use of AuNP functionalized octahedral Cu_2_O nanocrystals and hemin/G-quadruplex tagged Cu_2_O-Au-BSA [70].

**Figure 6 biosensors-12-00767-f006:**
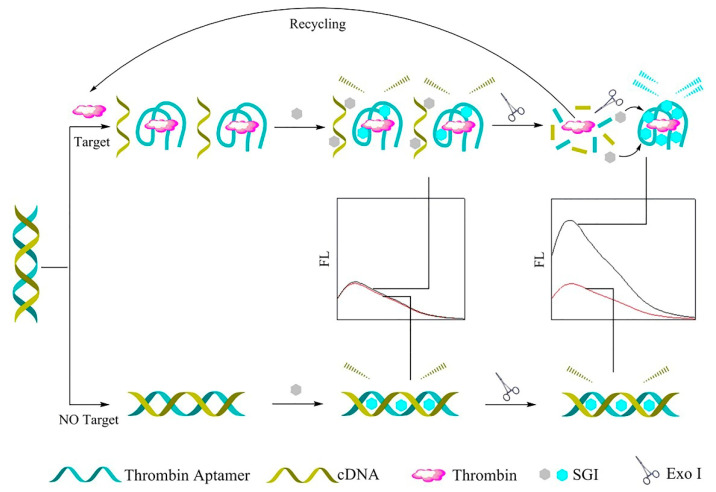
The label-free fluorescence aptasensor for TB identification [105].

**Figure 7 biosensors-12-00767-f007:**
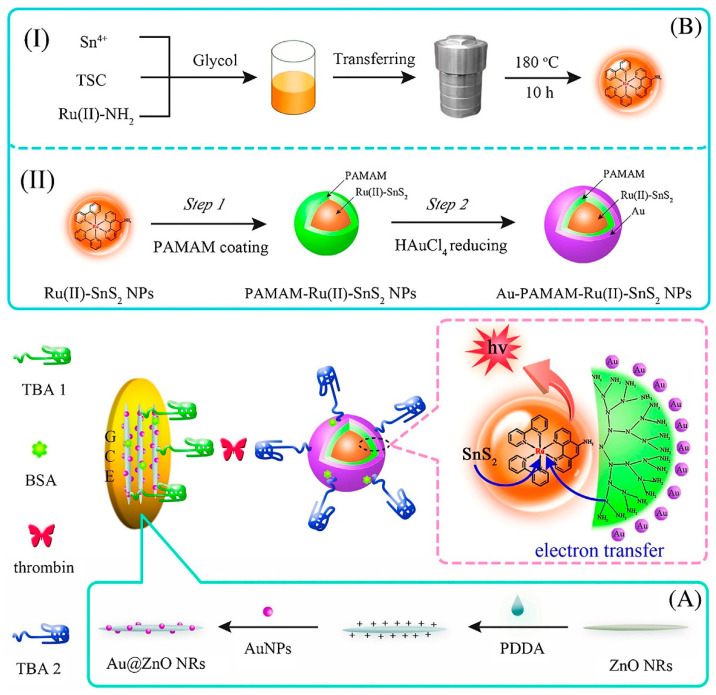
The electrochemiluminiscence-based aptamer for TB determination. The preparation process for TB sensing. (**A**) Schematic of the manufacturing of Au@ZnO NRs, and (**B**) self-enhanced tri-layer Au-PAMAM-Ru(II)-SnS2 nanoparticles’ gradual production process [121].

**Figure 8 biosensors-12-00767-f008:**
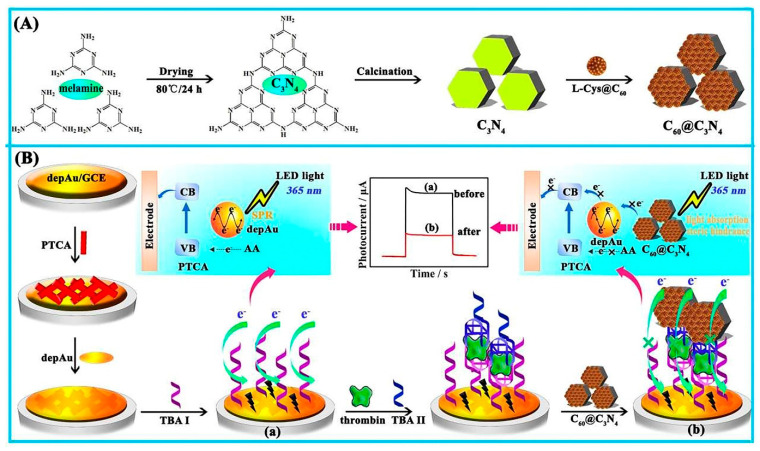
The photoelectrochemical aptasensor for TB determination. (**A**) The synthesis of C_60_@C_3_N_4_ nanocomposites, and (**B**) the signal-off PEC aptasensor fabrication mechanism [128].

**Figure 9 biosensors-12-00767-f009:**
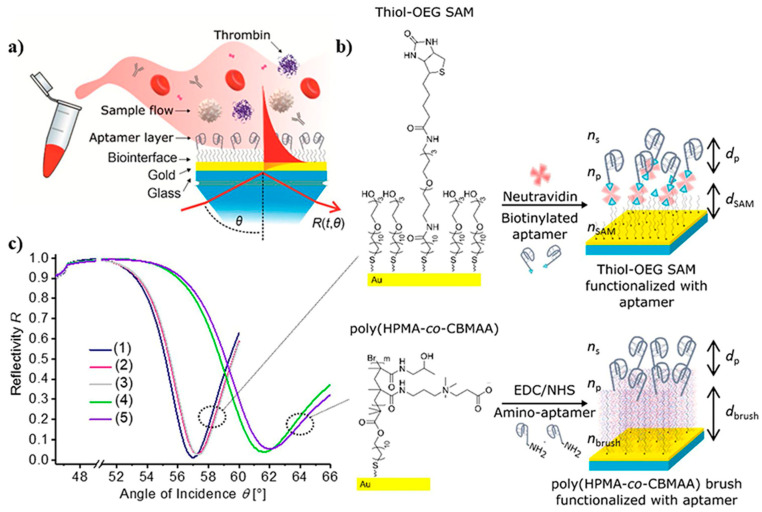
Surface plasmon resonance-based aptasensor for TB determination: (**a**) SPR sensor chip, (**b**) architectural design of thiol-OEG SAM or poly(HPMA-co-CBMAA) architecture and functionalized with aptamer, and (**c**) reflection at an angle of SPR spectra obtained at a specific wavelength on an SPR sensor chip includes (1) thiol-OEGSAM, (2) thiol-OEG SAM with neutravidin, (3) a comparison of spectra (1 and 2) with (4 and 5), (4) pristine polymer brushes, and (5) polymer brushes functionalized with aptamer HD22 [132].

**Figure 10 biosensors-12-00767-f010:**
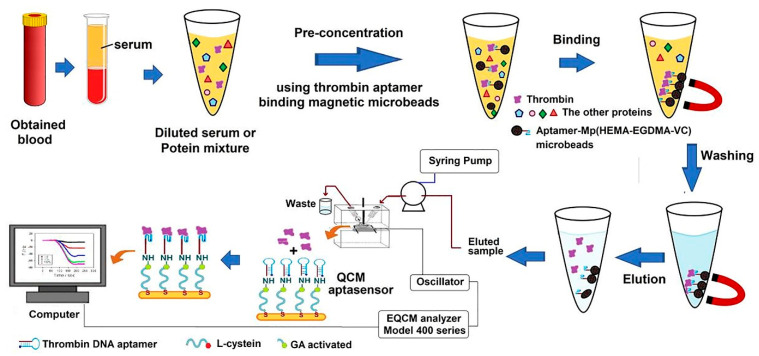
The process for TB determination by using a QCM-aptasensor [174].

**Table 1 biosensors-12-00767-t001:** The proposed electrochemical biosensors for the determination of TB.

Transduction Method	Signal Probe	Signal Amplification	Detection Limit	Linear Range	Ref
DPV	Tetraferrocene	Target-triggered CHA	0.28 pM	0.001–3.125 nM	[21]
DPV	MXene-Au-MB	Ferrocene (Fc)	16.1 fM	0.1 pM to 10 nM	[71]
SWV	CPAD and ferrocenylmethyl methacrylate	Ferrocene	0.062 pM	10−250 μU mL^−1^	[72]
DPV	A1-TB-A2 (binding-based DNA architecture)	[Ru(NH_3_)_6_]^3+^	32 fM	100 fM to10 nM	[73]
SWV	Au nanoparticles	Alkaline phosphatase (ALP)	0.26 fM	1 fM to 10 nM	[74]
DPV	Methylene blue (MB)	Pb^2+^-dependent DNAzyme	1.8 pM	5 pM to 5 nM	[26]
EIS	MoS_2_/PCN-223-Fe	PCN-223-Fe	0.03 pM	0.1 pM to 100 nM	[75]
DPV	ZnCr-LDH	Multi-walled carbon nanotubes (MWCNT)	0.1 fM	0.005 pM–12 nM	[76]
DPV	Conductive supramolecular polymer hydrogel (CSPH)	TBA1 and MNP-TBA2	0.64 pM	1 pM to10 nM	[29]
DPV	Methylene blue (MB)	MNPs	0.8 fM	10 fM to100 nM	[77]
DPV	Bisferrocene	CHA + bisferrocene	0.18 pM	0.25 pM–2.5 nM	[78]
CV and DPV	Cu_2_(CHDC)_2_	AuNPs	0.01 fM	0.2 to 1.0 mM	[79]
SWV	HAP nanoparticles and MNPs	Graphene	0.03 fM	0.1 fM to 1.0 nM	[80]
SWV	MB-DNA/Fc-DNA	AuNPs	56 fM	0.1 pM to 10 pM	[81]
DPV	Bisferrocene	Bisferrocene	0.8 pM	1.2 pM-12 nM	[82]
DPV	Tetraferrocene	Tetraferrocene	0.126 pM	18 pM–1.8 × 10 nM	[83]
DPV	Fc-PHNs	L-Cys	0.032 pM	0.1 pM to 80 nM	[84]
SWV	[Ru(NH_3_)_6_]^3+^	AuNPs	23.6 pM	0.05–100 nM	[85]
SWV	S1@PtNPs and S2@PtNPs	PtNPs and MnTMPyP	10.7 aM	1 fM to 100 nM	[86]
DPV	Methylene blue (MB)	_	0.57 fM	1 fM to 1 nM	[87]
SWV	TBA1-AgNP-GO	AgNP-GO	0.03 nM	0.05–5 nM	[88]
EIS	Reduced graphene oxide (rGO)	3, 4, 9, 10-perylenetetracarboxylic acid (PTCA)	0.2 pM	1 pM–100 nM	[89]
Amperometry	TBA 2–Au@ZIF-8(NiPd)	Au-COFs	15 fM	0.1 pM to 20 nM	[90]
DPV	Ferrocene and Azobenzene	_	3 pM	2.48 ± 0.02 and 20.26 ± 0.98 nM	[91]
DPV	Methylene blue	AuNPs coated ERGO nanosheets	0.17 nM	0.5 to 10 nM	[92]
DPV	Tetraferrocene	CHA + tetraferrocene	0.06 pM	0.12 pM–1.2nM	[93]
SWV	Methylene blue-modified Ot (MB-Ot)	RCA-CRISPR/Cas12a	1.26 fM	100 fM–10 nM	[94]
DPV	_	HCR	0.56 pM	1 pM–1 nM	[95]

**Table 2 biosensors-12-00767-t002:** The proposed optical biosensors for the determination of TB.

Transduction Method	Modified Materials	Detection Limit	Linear Range	Ref
ECL	3Dgraphene/Cu_2_OMWCNTs/RuSiNPs	1.3 × 10^−15^ M	5.0 × 10^−15^ to 5.0 × 10^−11^ M	[144]
ECL	EHNs/GCE	0.01 fM	0.01 fM–10 pM	[145]
ECL	RuAg/SiO_2_NPs@TBA II and PTG-AuNPs	1 fM	2 fM–2 pM	[146]
ECL	Ru(bpy)_3_^2+^ (RuND) and ferrocene	0.74 pM	1.0 × 10^−12^ to 1.0 × 10^−9^ M	[147]
ECL	3D graphene/Ru-PtNPs/cDNA/BSA/NGQDs@SiO_2_	23.1 fM	2.0 pM−50 nM	[148]
ECL	Ru(bpy)_3_^2+^/β-CD-AuNPs/Nafion/GCE	0.23 pM	0.4 to 1000 pM	[30]
Colorimetry	GOx-dHP and HRP-scCro	0.92 nM	0.5–10 nM	[149]
Colorimetry	AuNPs	4 nM	3.1–25 nM	[150]
Colorimetry	AuNP-PDA liposome	_	_	[151]
ECL	PAMAM-QDs and Au@Luminol	1.82 fM	10 fM to 1 nM	[152]
ECL	Ru-PEI-L-lys-ZIF-8 and PtNPs	0.02 aM	1 fM to 10 pM	[153]
ECL	HHTP-HATP-COF/S_2_O_8_^2−^	62.1 aM	100 aM to 1 nM	[154]
SPR	3,3′Dithiodipropionic acid di (N-hydroxysuccinimide ester) (DSP) and MCH	6.0 nM	30–100 nM	[155]
Fluorescent	CNNS/AgNCs	0.3 nM	1–800 nM	[156]
Fluorescent	DNA-AgNCs/PPyNPs	0.58 nM	2–40 nM	[157]
Fluorescent	DNA-AgNCs and GRS	300 pM	500 pM to 1600 nM	[158]
Fluorescent	smURFP and hydrophobin HGFI	0.2 aM	1.07 aM to 0.01 mM	[159]
PEC	ITO/TiO_2_/CQD	0.83 pM	1.0 to 250 pM	[160]
PEC	Au-rGO-CuS and CuInS2/b-TiO_2_	30 fM	0.1 pM to 10 nM	[161]
PEC	TiO_2_NTs/CuOx/PTCA/Pt	55 fM	0.0003 nM to 10 nM	[162]
PEC	Au–Ag_2_S	0.67 pM	1.0 to 10.0 pM	[163]
PEC	Perylene-3,4,9,10-tetracarboxylic acid (PTCA) and ferrocene	0.17 fM	0.5 fM–100 nM	[164]
PEC	P5FIn/PEDOT/ITO and aptamer/erGO/ITO	0.041 pM	0.10 × 10^−3^ nM–10.0 nM	[165]
Plasmonic plastic optical fiber	Gold and poly ethylene glycol (PEG)	1 nM	1.6–60 nM	[166]
Fluorescent	Fluorescein amidite (FAM) and Ti_3_C_2_ MXene	5.27 pM	20–200 pM	[167]
Fluorescent	stilbene	0.205 µM	0.01 to 2.5 µM	[168]
ECL	Ru(bpy)_3_^2+^ in 1-ethyl-3-methylimidazolium Tetrafluoroborate (EMImBF4)	0.74 pM	1.0 × 10^−12^ to 1.0 × 10^−9^ M	[169]
PEC	TNA/g-C_3_N_4_	3.4 fM	0.01–500 pM	[170]
ECL	f1-TiO_2_/g-C_3_N_4_/PDA	8.9 × 10^−12^ M	1.0 × 10^−11^ to 1.0 × 10^−5^ M	[171]
luminescence resonance energy transfer (LRET)	Ag_2_Se QDs	0.034 nM	0.1 nM to 125 nM	[172]

## Data Availability

Not applicable.

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
