# Peer review of "Recent Progresses in Development of Biosensors for Thrombin Detection"

_biosensors, 2022, doi:10.3390/bios12090767_

Round 1
Reviewer 1 Report
1. Two tables summarizing and comparing the DNAzymes-Based aptasensors and RNAzymes-Based aptasensors are recommended.
2. Pls refer to the article published recently reviewing the DNAzymes-Based aptasensors for TB detection for more information.
The Recent Progress in DNAzymes-Based Aptasensors for Thrombin Detection. Crit Rev Anal Chem. 2022 Jul 22;1-22.
3. Pls discuss the potential application of 3D printing technology with the biosensor with the following papers. ACS Biomaterials Science & Engineering 7 (9), 4009-4026, Advanced Materials, 2108931; Advanced Materials 34 (1), 2107038.
4. Pls refer to the article published recently reviewing the electrochemical aptasensors for TB detection for more references.
Nanomaterial-Based Label-Free Electrochemical Aptasensors for the Detection of Thrombin. Biosensors (Basel). 2022 Apr 16;12(4):253.
5. Pls carefully check the format of your references. For example, Line 178 had an error in reference source.
6. Did you get the permission for reproducing the cited figures?
7. There are two Table 1 in the manuscript and I believe the second table 1 should be table 2.
8. Pls cite the following paper “The Journal of arthroplasty 33 (3), 786-793”. to discuss the relation between surgery and the thrombosis.
Author Response
Reviewer: 1
Comments:
- Two tables summarizing and comparing the DNAzymes-Based aptasensors and RNAzymes-Based aptasensors are recommended.
Response: We appreciate the reviewer’s suggestion. Regarding your point, we investigated RNAzymes-Based aptasensors, but due to a lack of relevant articles, we were unable to provide more tables and information. But DNAzymes-Based aptasensors were explained in introduction section.
- Pls refer to the article published recently reviewing the DNAzymes-Based aptasensors for TB detection for more information. (The Recent Progress in DNAzymes-Based Aptasensors for Thrombin Detection. Crit Rev Anal Chem. 2022 Jul 22;1-22.)
Response: Thanks for your useful proposed article. We cited and used this article, and have now stated these further relevant points in the manuscript:
" Electrochemical biosensors are electrode-dependent sensors that measure the output of interactions of biomolecules with their targets on an electrode surface." (Page 3 Line 124-125)
- Pls discuss the potential application of 3D printing technology with the biosensor with the following papers. ACS Biomaterials Science & Engineering 7 (9), 4009-4026, Advanced Materials, 2108931; Advanced Materials 34 (1), 2107038.
Response: Your suggestions and proposed articles were useful for improving the context. We appreciate the suggestion. We cited these papers and have included a few sentences as the following:
" Moreover, many 3D printing technologies have been recently utilized to construct biosensors or some parts of them due to the benefits of these approaches over conventional ones, such as end-user customization, great functionality and adaptability, and quick prototyping. This technology has also quickly been developed in a wide range of biological and biomedical fields, including organ on a-chip platforms, bioelectronics, microfluidic devices, and tissue-engineered implants." (Page 2 Line 57-63)
- Pls refer to the article published recently reviewing the electrochemical aptasensors for TB detection for more references. (Nanomaterial-Based Label-Free Electrochemical Aptasensors for the Detection of Thrombin. Biosensors (Basel). 2022 Apr 16;12(4):253.)
Response: Thanks for your useful proposed article. We used this article and have now stated these further relevant points in the manuscript:
"Although various electrochemical biosensors for TB measurement have been developed, there are still several constraints that must be resolved before their transfer into real-world applications. For instance, starting blood coagulation cascades by contact of the electrode as a foreign body with blood, which causes biofouling of the electrodes [42]. In addition, as regards to the presence of thrombin in nM quantities in the blood, there is a requirement for very sensitive biosensors for efficient detection. Utilization of nanoparticles and nanostructures can theoretically overcome these concerns and problems [38]." (Page 3 Line 132-138)
- Pls carefully check the format of your references. For example, Line 178 had an error in reference source.
Response: Thank you for your attention. The related reference has been revised.
- Did you get the permission for reproducing the cited figures?
Response: Thank you for your attention and pointing this out. We got permission to use the figures in our manuscript, and related documents were submitted previously together with the manuscript file.
- There are two Table 1 in the manuscript and I believe the second table 1 should be table 2.
Response: Thank you for your attention. We are not sure about this comment since as regards to the manuscript, there are two tables with captions "Table 1" and "Table 2", respectively. So we did not make any changes to that.
- Pls cite the following paper “The Journal of arthroplasty 33 (3), 786-793”. to discuss the relation between surgery and the thrombosis.
Response: Your suggestion and proposed article was useful for improving the content. We appreciate the suggestion. We cited the intended paper and have included a few sentences as the following:
" Deep vein thrombosis is the generation of blood clots in the deep veins. Surgery, turbulence of blood flow, and endothelial injury are the factors that predispose to it, which surgery is the most common reason. It is a dangerous disorder that can cause morbidity and mortality." (Page 2 Line 87-90)
Reviewer 2 Report
The author presents about the development of biosensors for detection of the thrombin. The manuscripts lack a connection of an ideas. The author discusses about the biosensor and connect with a transducer, then again to aptasensor. The advantages of aptasensor are already found in the literature and there is no new novelty presented in the review paper.
1. In the end of first paragraph, the author talked about the modern platforms for the detection of proteins has been developed, but they have not presented which methods are they refereeing too.
2. How does the thrombin associate with the metastasis?
3. How does this thrombin together with the shear stress effects the survivability of circulating tumour cells? The author might want to present these ideas in the initial discussion part.
4. The authors talk about the thrombin and metastasis and suddenly to what happens when TB comes to vascular injuries. Then to the concentration of TB in the blood. The flow of the content is not great and lacks the connections of an ideas.
5. In the last paragraph of the introduction, the author talk about the several types of biosensors have been developed for detecting TB but do not mention about the name of the biosensors.
6. The discussion of electrochemical biosensors is too basic and can be found in the literature already.
7. How are the voltametric electrochemical biosensors used in the measurement of TB?
8. Please check reference 30. It shoes (Error! Reference source not found.Figure 2) [30].
9. The author talked about the strategies to analyze TB in blood samples using an allosteric kissing complex-based electrochemical biosensor. How is this used to detect TB?
10. What is kissing complex refer to?
11. Can the graphene quantum dots presented here (J Ju, S Regmi, A Fu, S Lim, Q Liu Journal of Biophotonics 12 (6), e201800367) be used in measure TB in blood samples? What is the author’s opinion on it?
12. Please see line 393. Error! Reference source not found.
13. In the Figure 9. Surface Plasmon Resonance-based aptasensor for TB determination, has the author drawn figure by themselves or taken from the specific paper? If they have taken by themselves, that is fine. Otherwise, please use the permission to use it.
14. There is already a recent review of biosensors for thrombin detection as presented in article below:
Sun, Hongzhi, Nannan Wang, Lin Zhang, Hongmin Meng, and Zhaohui Li. "Aptamer-Based Sensors for Thrombin Detection Application." Chemosensors 10, no. 7 (2022): 255.
Salmasi, Zahra, Nadiyeh Rouhi, Hossein Safarpour, Nozhat Zebardast, and Hamed Zare. "The Recent Progress in DNAzymes-Based Aptasensors for Thrombin Detection." Critical Reviews in Analytical Chemistry (2022): 1-22.
Hence, I do not see the author innovation to establish their own biosensors for specific sensitive detection of target analytes and hence I would not recommend this review to publish in the biosensor journal.
Author Response
Reviewer: 2
Comments:
- In the end of first paragraph, the author talked about the modern platforms for the detection of proteins has been developed, but they have not presented which methods are they refereeing too.
Response: Thank you for pointing this out. We have now revised it as the following:
" The clinical methods for the detection of proteins are usually related to immunoassays, which contain some restrictions. Therefore, modern platforms for the detection of proteins have been developed. For this purpose, recently, aptamers and DNAzymes have been used for bioanalysis, nanotechnology, clinical diagnostics and therapy [16]. Moreover, many 3D printing technologies have been recently utilized to construct biosensors or some parts of them due to the benefits of these approaches over conventional ones, such as end-user customization, great functionality and adaptability, and quick prototyping. This technology has also quickly been developed in a wide range of biological and biomedical fields, including organ on a-chip platforms, bioelectronics [17,18], microfluidic devices [19], and tissue-engineered implants [20]." (Page 2 Line 53-63)
- How does the thrombin associate with the metastasis?
Response: The definition of metastasis is "the spread of cancer cells from the area where they initially developed to another region of the body." Thrombin promotes cancer development and metastasis through platelets. Thrombin activates platelets and enhances their interactions with tumor cells, which have a key role in tumor development and metastasis. Platelets promote tumorigenesis through a variety of mechanisms, including the secretion of factors that promote tumor cell proliferation, pro-angiogenic factors, and cellular expression alterations. In addition, they also interact directly with endothelial cells to promote angiogenesis. Platelet and fibrin aggregates form around circulating tumor cells, and protecting them from shear forces and host immunity.
(Reddel, C.J., Tan, C.W. and Chen, V.M., 2019. Thrombin generation and cancer: Contributors and consequences. Cancers, 11(1), p.100.)
- How does this thrombin together with the shear stress effects the survivability of circulating tumour cells? The author might want to present these ideas in the initial discussion part.
Response: We appreciate the suggestion. Thrombin activates platelets and enhances their interactions with tumor cells, which have a key role in tumor development and metastasis. Platelet and fibrin aggregates form around circulating tumor cells, and protecting them from shear forces and host immunity.
We have included a few sentences based on the suggestion as the following in the manuscript:
"TB promotes tumor cell adherence to platelets, endothelial cells, and subendothelial matrix proteins. Platelet-mediated tumor cell sequestration shields tumor cells from immunologic host surveillance by preventing tumor cell eradication by natural killer cells, and so prolongs tumor survival in the circulation. Platelets also increase tumor cell growth and angiogenesis, as well as metastasis." (Page 2 Line 94-98)
- The authors talk about the thrombin and metastasis and suddenly to what happens when TB comes to vascular injuries. Then to the concentration of TB in the blood. The flow of the content is not great and lacks the connections of an ideas.
Response: Thanks for your useful comments. We apologize for the lack of coherence and cohesion. Your point is absolutely correct, and the relevant text was reviewed and revised to resolve this issue. We also added several sentences based on the suggestions of reviewers to provide better content. We have now revised it as the following:
" Thrombin (TB) is a serine protease and has a key function in pathological and physiological coagulation, blood vessel hemostasis, and wound healing [21]. It is a special molecule that acts as both a procoagulant and an anticoagulant. TB controls its generation through activating coagulation factors V, VIII, and XI. When it comes to vascular injuries, it is generated from prothrombin in the wound area and induces the conversion of fibrinogen to fibrin. It also stimulates platelets, and through the activation of the factors XI and XIII, respectively, prevents fibrinolysis of the fibrin clots and affects the formation of a firm fibrin clot. This process, in turn, forms blood clots and, in such a way, prevents bleeding. The coagulation procedure begins when the concentration of TB in the blood reaches around 5–20 nM, and its concentration after the clot formation can be several hundred nM. The activity of TB as an anticoagulant is performed through binding to thrombomodulin (a membrane receptor protein), which begins a chain of actions that results in fibrinolysis [22,23]. The generation of TB is categorized into three sequential stages based on its final concentration and other relevant factors: i) initiation (TB concentration is from 1 nM to 20–30 nM), ii) propagation (from 20–30 nM up to more than 800 nM), and iii) termination (when TB production stops and free thrombin is passive) [24]. Generally, the concentration of TB in the blood depends on the person’s physical condition. It may not be found in healthy people's blood, while its level may vary from nM to µM during the coagulation cascade [25]. The Imbalances in its concentration can lead to various diseases such as venous thromboembolism, nephrotic progress, central nervous system diseases, etc. In a specific case, pregnant women, who experience a physiological hypercoagulable state, are in the peril of deep vein thrombosis and sets of momentous gestation complications like preeclampsia, recurrent pregnancy loss, intrauterine growth restriction, or intra-uterine fetal death [26]. Deep vein thrombosis is the generation of blood clots in the deep veins. Surgery, turbulence of blood flow, and endothelial injury are the factors that predispose to it, which surgery is the most common reason. It is a dangerous disorder that can cause morbidity and mortality [27,28]. Moreover, the alteration in the levels of TB is associated with leukemia, vascular wall inflammation, tumor growth, and metastasis [29]. As an illustration, it has a modulating effect on tumor cell proliferation by enhancing growth at a low concentration of cancer cells and preventing growth at a high concentration and can also produce an apoptosis effect [30]. TB promotes tumor cell adherence to platelets, endothelial cells, and subendothelial matrix proteins. Platelet-mediated tumor cell sequestration shields tumor cells from immunologic host surveillance by preventing tumor cell eradication by natural killer cells, and so prolongs tumor survival in the circulation. Plate-lets also increase tumor cell growth and angiogenesis, as well as metastasis [31]." (Page 2 Line 64-99)
- In the last paragraph of the introduction, the authors talk about the several types of biosensors have been developed for detecting TB but do not mention about the name of the biosensors.
Response: Thank you for your constructive comment. We have now revised it as the following: " Thus far, several types of biosensors have been developed for detecting TB, namely enzyme-based, immunosensors, DNA-based biosensors, tissue-based, and thermal and piezoelectric biosensors." (Page 3 Line 108-110)
- The discussion of electrochemical biosensors is too basic and can be found in the literature already.
Response: We sincerely appreciate this constructive comment. We apologize for the lack of enough information. We acknowledge this fact pointed out by the reviewer and we added some further information:
- "They have made significant advancements in clinical diagnosis, food quality regulation, and environmental monitoring." (Page 3 Line 120-122)
- " Electrochemical techniques are favorable because of their great sensitivity, quick response, tiny sample consumption, and low cost." (Page 3 Line 122-123)
- " Electrochemical biosensors are electrode-dependent sensors that measure the output of interactions of biomolecules with their targets on an electrode surface. Typically, electrochemical sensing employs a transducer and three electrodes (counter electrode, working electrode, and reference electrode)." (Page 3 Line 123-127)
- " Various electrochemistry-driven biosensing techniques have recently been proposed for cost-effective and miniaturized analytical instruments based on amperometry, voltammetry, potentiometry, and electrochemical impedance spectroscopy (EIS). Although various electrochemical biosensors for TB measurement have been developed, there are still sever-al constraints that must be resolved before their transfer into real-world applications. For instance, starting blood coagulation cascades by contact of the electrode as a foreign body with blood, which causes biofouling of the electrodes. In addition, as regards to the presence of thrombin in nM quantities in the blood, there is a requirement for very sensitive biosensors for efficient detection. Utilization of nanoparticles and nanostructures can theoretically overcome these concerns and problems." (Page 3 Line 129-138)
- How are the voltametric electrochemical biosensors used in the measurement of TB?
Response: Thank you for pointing this out. Voltammetric techniques use a time-dependent changeable voltage with respect to the reference electrode to quantify the current response between the working electrode and the counter electrode in the region of the conducting of oxidation-reduction reactions. The three-electrode system is easily constructed on a single substrate and enables measurement of a tiny amount of sample.
We have also added this information to the manuscript. (Page 3 Line 143-147)
- Please check reference 30. It shoes (Error! Reference source not found.Figure 2) [30].
Response: Thank you for your attention. The related reference has been revised.
- The author talked about the strategies to analyze TB in blood samples using an allosteric kissing complex-based electrochemical biosensor. How is this used to detect TB?
Response: We thank for this opportunity to clarify this point. In this research, after forming the kissing complex, an electrochemical response that has been produced by applying streptavidin-labeled alkaline phosphatase to the electrode decreases with the linking of the thrombin to the recognition regions on the HP2 scaffold. In this way, the discrepancy between the electrochemical signal before and after TB binding could be used to efficiently evaluate its recognition.
We have revised the sentences in the manuscript to facilitate better reading and understanding of this section as the following:
" Zhao et al. designed an allosteric kissing complex-based electrochemical biosensor to detect TB. This method used the apical loop-loop or kissing interaction of the RNA-RNA base sequences to generate a kissing complex with two designed hairpins (immobilized HP1 and HP2 scaffold) on the electrode surface. After forming the kissing complex, an electrochemical response was produced by applying streptavidin-labeled alkaline phosphatase to the electrode. When TB, as the target protein, was bound to the recognition regions linked onto the HP2 scaffold, the HP2 stem unfolded due to the steric strain, which resulted in a reduction in electrochemical signal relevant to protein quantification. In this way, the discrepancy between the electrochemical signal before and after TB binding could be used to efficiently evaluate its recognition." (Page 7 Line 279-288)
- What is kissing complex refer to?
Response: In this research, an apical loop-loop or kissing interaction of the RNA-RNA base sequences has been used to generate a kissing complex with two designed hairpins (HP1 and HP2 scaffold). Hp1 has been immobilized on the electrode surface, which could form a kissing complex with Hp2 through the apical loop-loop or kissing interaction of the RNA-RNA base sequences. The Hp2 possesses appended single-stranded tails on each end, which hybridize with the recognition element-conjugated DNA strands to construct a protein-responsive switch of Hp2 scaffold.
- Can the graphene quantum dots presented here (J Ju, S Regmi, A Fu, S Lim, Q Liu Journal of Biophotonics 12 (6), e201800367) be used in measure TB in blood samples? What is the author’s opinion on it?
Response: Regarding your proposed paper, we did a search for graphene quantum dot based charge-reversal. There is no article about the application of graphene quantum dot based charge-reversal for TB biosensors to date. However, as mentioned in the manuscript, graphene quantum dots have been utilized in TB biosensors.
- Please see line 393. Error! Reference source not found.
Response: Thank you for your attention. The related reference has been revised.
- In the Figure 9. Surface Plasmon Resonance-based aptasensor for TB determination, has the author drawn figure by themselves or taken from the specific paper? If they have taken by themselves, that is fine. Otherwise, please use the permission to use it.
Response: Thank you for your attention and pointing this out. We got permission to use the figures in our manuscript, and related documents were submitted previously together with the manuscript file.
- There is already a recent review of biosensors for thrombin detection as presented in article below:
Sun, Hongzhi, Nannan Wang, Lin Zhang, Hongmin Meng, and Zhaohui Li. "Aptamer-Based Sensors for Thrombin Detection Application." Chemosensors 10, no. 7 (2022): 255.
Salmasi, Zahra, Nadiyeh Rouhi, Hossein Safarpour, Nozhat Zebardast, and Hamed Zare. "The Recent Progress in DNAzymes-Based Aptasensors for Thrombin Detection." Critical Reviews in Analytical Chemistry (2022): 1-22.
Response: Thank you for your attention. Due to the fact that existing reviews have been discussing a specific issue about thrombin biosensors, we have made an attempt to address all types of recent proposed TB biosensors and make a comprehensive review. In the first review, " Aptamer-based sensors for thrombin detection application", only the proposed aptasensors for thrombin detection were covered. Also, only the DNAzymes-based aptasensors have been considered in the second review, "The recent progress in DNAzymes-based aptasensors for thrombin detection", whereas we attempted to present a comprehensive study that included all types of recent thrombin detection biosensors in addition to aptasensors.
Reviewer 3 Report
The current manuscript entitled “ Recent progresses in development of biosensors for thrombin detection” by “Eivazzadeh-Keihan et al” deliberated on the various biosensors proposed in the literature which have been designed for the thrombin detection. According to their various transducers, constructions, and compositions, the performance, benefits, and restrictions of each are summarized and compared. The manuscript seems good and well written. The review will be useful for the scientific readers. The work can be accepted after addressing the following comments.
1. Introduction seems less and poorly written, still there is scope to increase the content of the work in the introduction section. The authors can discuss about the various on-site sensing strategies for the detection of different targets or analytes. For example, please see the following article.
https://doi.org/10.1016/j.ccr.2021.214305
Note: The article is given, for your reference only. The authors no need to cite this article.
2. In the introduction compare the existing reviews with the current review.
3. Provide the advantages of electrochemical sensing strategies.
4. Among the proposed sensing strategies, which sensors have good sensing performance.
5. The authors need to provide effective challenges in the section 5. Seems section 5 is not sufficient.
6. Conclusion should be written as conclusions.
7. Detailed conclusions should be presented.
Author Response
Reviewer: 3
Comments:
The current manuscript entitled “Recent progresses in development of biosensors for thrombin detection” by “Eivazzadeh-Keihan et al” deliberated on the various biosensors proposed in the literature which have been designed for the thrombin detection. According to their various transducers, constructions, and compositions, the performance, benefits, and restrictions of each are summarized and compared. The manuscript seems good and well written. The review will be useful for the scientific readers. The work can be accepted after addressing the following comments.
We thank the reviewer for kind words.
- Introduction seems less and poorly written, still there is scope to increase the content of the work in the introduction section. The authors can discuss about the various on-site sensing strategies for the detection of different targets or analytes. For example, please see the following article.
https://doi.org/10.1016/j.ccr.2021.214305
Response: We sincerely appreciate this important and constructive point. Your point is absolutely correct. We have added several sentences and modified existing sentences to include more information and strengthen the introduction section. Your proposed article was also useful for improving the content. We appreciate the suggestion. We have now stated this further relevant information in the manuscript:
- " A biosensor is a device that generally consists of three components, containing a biological element, a transducer, and a signal processor." (Page 1 Line 36-37)
- " Biosensing technology has advanced in recent decades, with improvements in biorecognition, transducers, and signal processing. The biological element can be an enzyme, anti-body, protein, whole cell, or aptamer. The transducer is the main component of the bio-sensors, and they can be classified into optical, electrochemical, piezoelectrical, calorimetric, or thermal categories, based on the physico-chemical characteristics of the transducer." (Page 1 Line 37-42)
- " Subsequently, the transducer converts the connection event into recognizable and quantifiable signals. An electrical instrument or an observer can be employed to monitor this signal." (Page 1 Line 44-46)
- " Indeed, the recognition element in aptasensors is a DNA or RNA aptamer. Aptasensors are a kind of biosensor that can be used in a wide range of experimental conditions for diagnostics and drug delivery since they offer many advantages, such as low-cost, quick response, easier modification, and easy merging within devices." (Page 2 Line 47-51)
- " For this purpose, recently, aptamers and DNAzymes have been used for bioanalysis, nanotechnology, clinical diagnostics and therapy. Moreover, many 3D printing technologies have been recently utilized to construct biosensors or some parts of them due to the benefits of these approaches over conventional ones, such as end-user customization, great functionality and adaptability, and quick prototyping. This technology has also quickly been developed in a wide range of biological and biomedical fields, including organ on a-chip platforms, bioelectronics, microfluidic devices, and tissue-engineered implants." (Page 2 Line 56-63)
- " Thrombin (TB) is a serine protease and has a key function in pathological and physiological coagulation, blood vessel hemostasis, and wound healing. It is a special molecule that acts as both a procoagulant and an anticoagulant. TB controls its generation through activating coagulation factors V, VIII, and XI." (Page 2 Line 64-67)
- " It also stimulates platelets, and through the activation of the factors XI and XIII, respectively, prevents fibrinolysis of the fibrin clots and affects the formation of a firm fibrin clot." (Page 2 Line 69-71)
- " The activity of TB as an anticoagulant is performed through binding to thrombomodulin (a membrane receptor protein), which begins a chain of actions that results in fibrinolysis." (Page 2 Line 74-76)
- " Deep vein thrombosis is the generation of blood clots in the deep veins. Surgery, turbulence of blood flow, and endothelial injury are the factors that predispose to it, which surgery is the most common reason. It is a dangerous disorder that can cause morbidity and mortality." (Page 2 Line 87-90)
- " TB promotes tumor cell adherence to platelets, endothelial cells, and subendothelial matrix proteins. Platelet-mediated tumor cell sequestration shields tumor cells from immunologic host surveillance by preventing tumor cell eradication by natural killer cells, and so prolongs tumor survival in the circulation. Platelets also increase tumor cell growth and angiogenesis, as well as metastasis." (Page 2 Line 94-99)
- " Hence, TB is employed as a main criterion for coagulation disorders, and accurate evaluation of its level assists in determining the course of diseases and developing treatment plans." (Page 3 Line 101-103)
- " Thus far, several types of biosensors have been developed for detecting TB, namely enzyme-based, immunosensors, DNA-based biosensors, tissue-based, and thermal and piezoelectric biosensors. This comprehensive overview incorporates developments in biosensors for effective TB detection, which have been classified based on the transducer as optical, electrochemical, and other ones, and is the first systematic review that almost addresses all types of recent proposed TB biosensors." (Page 3 Line 108-113)
- In the introduction compare the existing reviews with the current review.
Response: Thank you for this comment. Due to the fact that existing reviews have been discussing a specific issue about thrombin biosensors, we have made an attempt to address all types of recent proposed TB biosensors and make a comprehensive review. We have added a few sentences on this in the last paragraph of the introduction section based on the suggestion. To clarify this point, we have now noted: " This comprehensive overview incorporates developments in biosensors for effective TB detection, which have been classified based on the transducer as optical, electrochemical, and other ones, and is the first systematic review that almost addresses all types of recent proposed TB biosensors. The present literature update has considered the last seven years of published literature. It provides an outlook on the current technologies in TB biosensors and helps acknowledge deficiencies in this field to be covered in future research." (Page 3 Line 110-116)
- Provide the advantages of electrochemical sensing strategies.
Response: Thank you for pointing this out. Information has been added to the manuscript in the " first paragraph of the Electrochemical biosensors" section. we have now noted: " Electrochemical techniques are favorable because of their great sensitivity, quick response, tiny sample consumption, and low cost." (Page 3 Line 122-123)
- Among the proposed sensing strategies, which sensors have good sensing performance.
Response: This is a controversial issue. As you know, the most important factors which we can consider to evaluate the sensors' performance are sensitivity, limit of detection (LOD), selectivity, and response time. These factors are important in the design of thrombin biosensors to obtain optimal sensing performance. Due to the various proposed sensing strategies, which have generally been classified into electrochemical and optical techniques in our manuscript, we can consider these factors separately. In terms of the LOD factor, which is defined as the lowest concentration that can be reliably detected, almost the electrochemical based biosensors and among them, the Voltammetric techniques, had the best results. However, in terms of time response factor, the optical methods possessed a rather faster response time. If we consider all of these, we can't generally note one strategy that has the best performance.
- The authors need to provide effective challenges in the section 5. Seems section 5 is not sufficient.
Response: Thank you for your constructive comment. We have expanded the Future trends and conclusions part and noted the following additional relevant points in the manuscript:
- " For instance, imbalancement of its level in the blood causes a wide range of abnormalities, including cancer, venous thromboembolic diseases, nephrotic progression, nervous system disorders, chronic inflammatory diseases, and other diseases." (Page 28 Line 1042-1044)
- " Although numerous biosensors have been introduced to date for the detection of thrombin, there are several challenges involved in their advancement. Some of these challenges include: i) improving the transduction process; ii) biofouling of the electrodes; iii) biosensor miniaturization, in which miniaturized nano-aptasensors can be sophisticated sensing platforms due to the development of microfluidic and on-chip-based techniques; iv) increasing transducer performance by improving sensitivity, selectivity, reproducibility, shortening response time, etc; v) the presence of thrombin at nM levels and alteration of its content in the human body by physiological processes show a demand for the development of highly sensitive biosensors to monitor thrombin content in vivo. As regards to experience, these challenges can be overcome by combining sensor technologies with nanomaterials, and it is noteworthy that optimal performance has been attained by the combination of multi-material components. Future studies need to concentrate on integrating functional nanomaterials to enhance synergistic effects and therefore improve the sensor's overall performance. Meanwhile, increasing the characteristics of the proposed biosensors, together with improving the biosensor's surface with various detection elements, can be employed for the simultaneous determination of TB and other important proteins in physiological samples." (Page 29 Line 1050-1066)
- Conclusion should be written as conclusions.
Response: Thank you for this comment. Correction has been made.
- Detailed conclusions should be presented.
Response: Thank you for your constructive comments. We have included a few sentences to provide more information in the conclusions section, which was noted in comment 5. Please refer to comment 5.
Round 2
Reviewer 2 Report
There are already enough literature in the field for the use of biosensors for the detection of thrombin.
https://www.mdpi.com/2227-9040/10/7/255
https://www.sciencedirect.com/science/article/pii/S0039914022002806?casa_token=VyL7gdgJHgsAAAAA:9l4WUZTTRsgXmOKyXvAIYbOgQ6UBSkFkXM1_Tb3D1ylUa2wosIxwxkrwN7rjRX0wP0wBEGkrQA
I do not see any new novelty in the paper and hence I would recommend this paper to reject.
